# Light Absorption by Brown Carbon over the South-East Atlantic Ocean

Lu Zhang[1], Michal Segal-Rozenhaimer[1,2], Haochi Che[1], Caroline Dang[3,4], Arthur J. Sedlacek III[5], Ernie R. Lewis[5], Amie Dobracki[6], Jenny P.S. Wong[7], Paola Formenti[8], Steven G. Howell[9], Athanasios Nenes[10,11]

[1]Department of Geophysics, Porter School of the Environment and Earth Sciences, Tel Aviv University, Tel Aviv, Israel

[2]Bay Area Environmental Research Institute, NASA Ames Research Center, Moffett Field, California, USA

[3]NASA Ames Research Center, Moffett Field, California, USA

[4]Oak Ridge Associated Universities, Oak Ridge, Tennessee, USA

[5]Brookhaven National Laboratory, Upton, New York, USA

[6]Rosenstiel School of Marine and Atmospheric Science, University of Miami, Miami, USA

[7]Mount Allison University, New Brunswick, CA

[8]Université Paris Cité and Univ Paris Est Creteil, CNRS, LISA, F-75013 Paris, France

[9]University of Hawaii at Manoa, Department of Oceanography, Honolulu, USA

[10]Laboratory of Atmospheric Processes and their Impacts, School of Architecture, Civil & Environmental Engineering, École Polytechnique Fédérale de Lausanne, Switzerland

[11]Center for Studies of Air Quality and Climate Change, Institute of Chemical Engineering Sciences, Foundation for Research and Technology Hellas, Greece

*Correspondence to*: Lu Zhang (luzhang@mail.tau.ac.il) and Michal Segal-Rozenhaimer (msegalro@tauex.tau.ac.il)

**Abstract.** Biomass burning emissions often contain brown carbon (BrC), which represents a large family of light-absorbing organics that are chemically complex, thus making it difficult to estimate their absorption of incoming solar radiation, resulting in large uncertainties in the estimation of the global direct radiative effect of aerosols. Here we investigate the contribution of BrC to the total light absorption of biomass burning aerosols over the South-East Atlantic Ocean with different optical models utilizing a suite of airborne measurements from the ORACLES 2018 campaign. An effective refractive index of black carbon (BC), $m_{eBC}=1.95+ik_{eBC}$, that characterizes the absorptivity of all absorbing components at 660 nm wavelength was introduced to facilitate the attribution of absorption at shorter wavelengths, i.e. 470 nm. Most values of the imaginary part of the effective refractive index, $k_{eBC}$, were larger than those commonly used for BC from biomass burning emissions, suggesting contributions from absorbers besides BC at 660 nm. The TEM-EDX single particle analysis further suggests that these long-wavelength absorbers might include iron oxides, as iron is found to be present only when large values of $k_{eBC}$ are derived. Using this effective BC refractive index, we find that the contribution of BrC to the total absorption at 470 nm ($R_{BrC,470}$) ranges from ~8-22 %, with the organic aerosol mass absorption coefficient

(MAC$_{OA,470}$) at this wavelength ranging from 0.30±0.27 to 0.68±0.08 m$^2$ g$^{-1}$. The core-shell model yielded much higher estimates of MAC$_{OA,470}$ and $R_{BrC,470}$ than homogeneous mixing models, underscoring the importance of model treatment. Absorption attribution using the Bruggeman mixing Mie model suggests a minor BrC contribution of 4 % at 530 nm, while its removal would triple the BrC contribution to the total absorption at 470 nm obtained using the AAE (absorption Ångström exponent) attribution method. Thus, it is recommended that the application of any optical properties-based attribution method use absorption coefficients at the longest possible wavelength to minimize the influence of BrC, and to account for potential contributions from other absorbing materials.

## 1 Introduction

Black carbon (BC) and brown carbon (BrC) are the two main light-absorbing carbonaceous aerosols that play a significant role in Earth's radiative forcing and climate (Bond et al., 2013; Laskin et al., 2015; Brown et al., 2018). BC is the principal atmospheric particulate absorber, and absorbs strongly over the entire solar spectrum (Bond and Bergstrom, 2006). Biomass burning (BB) contributes approximately 2/3 of the global primary organic aerosol (OA) budget, which is currently treated in most climate models as "white carbon" that only scatters and does not absorb light (Bond et al., 2013). However, studies show that BB OA contains a substantial amount of BrC, which predominantly absorbs at short visible and near-UV wavelengths (Chen and Bond, 2010; Lack et al., 2012; Saleh et al., 2014; Taylor et al., 2020). Feng et al. (2013) found that the absorption of BrC can shift BB direct radiative forcing to positive values. The contribution of BrC to the total absorption by carbonaceous aerosols is estimated to be ~20-50 % with a global radiative effect of ~0.03-0.6 W m$^{-2}$ (Kirchstetter and Thatcher, 2012; Wang et al., 2013; Feng et al., 2013; Saleh et al., 2015; Jo et al., 2016). However, values reported by these studies are subject to substantial uncertainties, and investigations of BrC properties are still very much in a developmental stage (Szopa et al., 2021). Thus, to realize a substantive reduction in the uncertainty, a better attribution of BrC light absorption is required.

Several methods have been employed to investigate the light absorption of BrC. The AAE (absorption Ångström exponent) attribution method utilizes the different spectral dependences of the absorption by BC and BrC to determine the contribution from BrC at short wavelengths (Lack and Langridge, 2013; Wang et al., 2018a; Taylor et al., 2020). Most earlier studies assumed the AAE of BC (AAE$_{BC}$) to be unity, although this assumption may introduce large uncertainties (Lack and Langridge, 2013) because AAE$_{BC}$ varies with size, mixing state, and wavelength (Lack and Cappa, 2010; Fig. 2 in Liu et al., 2015; Liu et al., 2018). Revised methods with more realistic values of AAE$_{BC}$ have been proposed, such as using the AAE from two long-visible wavelengths (Taylor et al., 2020) or that from the Mie model (Wang et al., 2018a). A major drawback of these methods is that they are suitable only for mixtures of BC and BrC, and if other absorbing materials are present, such as dusts, more

information is needed to account for their contribution to light absorption at longer wavelengths. Another methodology involves measuring the absorption of organics that have been extracted with either an organic solvent or water and represents, thus far, the only way to directly measure BrC absorption (Wong et al., 2019). The drawback to this approach is that not all organics can be extracted with one or two solvents, as highlighted by the work of Chen and Bond (2010), who reported extraction efficiencies of ~70 % in water and ~90 % in methanol. This inability to extract all organics means that insoluble organic substances remain unknown since they are not measured. This, in turn, could lead to cases where the absorption properties of the extracted organics might be different from those derived from in situ measurements. In addition, this method is carried out offline and requires elaborate laboratory analysis. Still another approach to estimate the absorption of BrC is through optical closure, determine the BrC absorption as the difference of the total measured absorption and that of BC calculated using Mie theory (Saleh et al., 2014; Liu et al., 2015). In this approach, the accuracy of BrC absorption relies heavily on the accuracy of BC absorption calculation. Values commonly used for the refractive index of BC ($m_{BC}$) from BB emissions in these calculations have ranged from 1.5+0.3i to 1.95+0.79i (Liu et al., 2015; Chylek et al., 2019; Taylor et al., 2020; Kahnert and Kanngießer, 2020), which will lead to large differences in BC absorption simulation results (Taylor et al., 2020). To date there is no consensus on the best value of $m_{BC}$. Another factor influencing the BC absorption calculation, and hence the estimated BrC absorption, is the mixing state of BC and non-BC components within particles. Liu et al. (2015) used the core-shell (CS) Mie model and Rayleigh-Debye-Gans approximation to investigate the effect of BC microphysics on the estimation of BrC absorption and found it to be highly sensitive to the model treatment. Saleh et al. (2014) compared internal and external mixtures of BC and BrC and found internally mixed cases yielded smaller BrC absorption than externally mixed ones. Similar to the AAE attribution method, the presence of absorbing materials other than BC and BrC can lead to errors in the attribution of BrC absorption. For example, light absorbing iron oxides (FeOx) was found to be common in field studies, especially for BB emissions (Ito et al., 2018), yet few studies perform measurements of size distributions and chemical composition of this particle type, making it difficult to separate their contribution to the total absorption from that of BrC.

The savannah regions in Africa experience widespread annual BB events from July to October, which are estimated to account to approximately 1/3 of global BB emissions (van der Werf et al., 2010). These aerosols are transported westward over the South-East Atlantic, making the region of the west coast of southern Africa and the South-Eastern Atlantic as that with the largest BrC absorption aerosol optical depth (Brown et al., 2018), and hence an ideal natural laboratory for investigating the absorption of BrC from BB emissions. In this study, we estimate the absorption of BrC using the optical closure method utilizing in-situ aircraft measurements and offline single particle analysis from the ORACLES (ObseRvations of Aerosols above CLouds and their intEractionS) 2018 campaign (Redemann et al., 2020). An effective refractive index of BC ($m_{eBC}$), which attempts to capture

the absorption of all possible absorbing components at 660 nm, is introduced to facilitate the absorption attribution at shorter wavelengths. The core-shell model and homogeneous models are applied and compared in this study. The range of values of the OA mass absorption coefficient ($MAC_{OA,470}$) and contribution of BrC to the total absorption at 470 nm ($R_{BrC,470}$) using the optical closure method are also obtained.

## 2 Methods

### 2.1 Site and Instrumentation

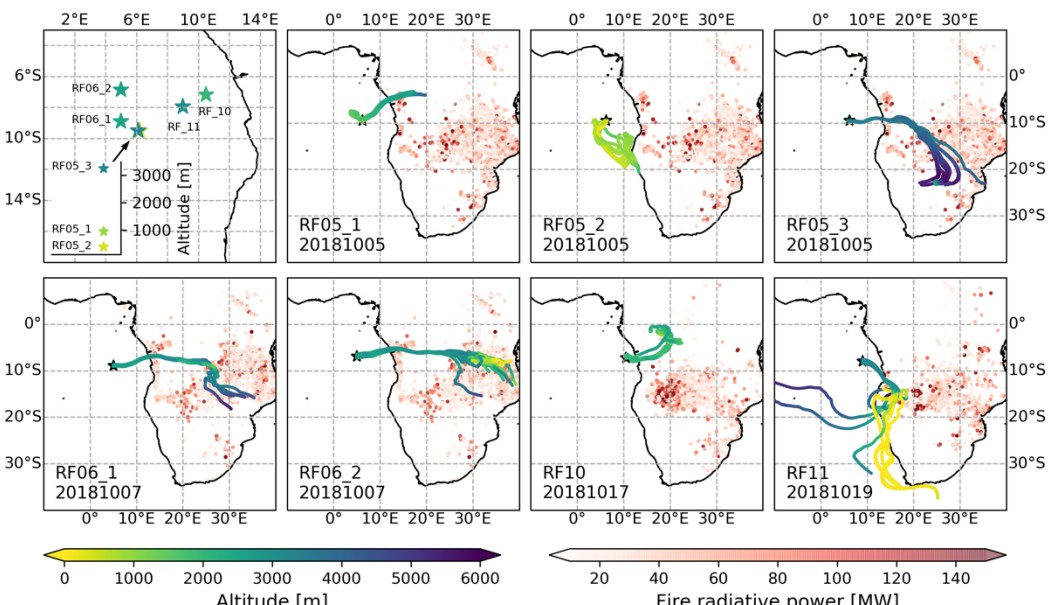

**Figure 1. (upper left panel) overview of the flight locations during ORACLES 2018 investigated in this study, and (all other panels) 7-day back trajectories and fire radiative power from MODIS 6 Collection Active Fire Detections (MCD14ML) at the end day of the trajectory (only data with confidence >50 % are used here).**

ORACLES was a three-year NASA-funded airborne field campaign to investigate the influence of BB emissions from southern Africa on regional and global climate (Redemann et al., 2020). We investigated aerosol optical properties from straight and level runs during seven research flights (RF) from the ORACLES 2018 campaign (Fig. 1): RF05_1, RF05_2, RF05_3, RF06_1, RF06_2, RF10, and RF11. The specifications for each flight can be found in Table 1 in Dang et al. (2021). These flights were chosen as they provided investigations of aerosol properties in the marine boundary layer (MBL) and free troposphere (FT), from relatively aged aerosols to highly aged aerosols, and are comprehensive in online and offline measurements with few missing data. RF05_1 and RF05_2 were within the MBL at an altitude of ~940 and 380 m, respectively; other flights were in the aerosol layer in the FT at an altitude equal to or greater than 2 km. The aerosol/plume age, modelled with a two-week forecast using the Weather Research and Aerosol Aware Microphysics (WRF-AAM) model

(Thompson and Eidhammer, 2014), was estimated to be ~11 days for RF05_1 and RF05_2, and ~7 days for RF05_3, RF06_1, RF06_2, RF10, and RF11 (Fig. S1).

The mass and number concentrations of the refractory BC particles and the single-particle mass and mixing state were determined by the Single-Particle Soot Photometer (SP2, Droplet Measurement Technologies). Coating thicknesses of rBC-containing particles were derived from scattering measurements using the leading-edge-only (LEO) method of Gao et al. (2007). The SP2 detects BC particles with core diameters from 80-650 nm; a lognormal function was fit to BC number size distribution to account for BC particles with core diameters outside this range (Schwarz et al., 2006). The mass equivalent diameter of BC was calculated as $(6m/\rho_{BC}\pi)^{1/3}$, where $m$ and $\rho_{BC}$ represent the mass and density of BC, respectively, with $\rho_{BC}$ assumed to be 1.8 g cm$^{-3}$ (Bond and Bergstrom, 2006). The uncertainty of the measured BC mass concentration and coating thicknesses are estimated to be 20 % and 22 %, respectively (Laborde et al., 2012). The mass ratio (MR) of non-BC substance to BC for BC-containing particles was determined by $\mathrm{MR} = \frac{\sum_i (D_{p,i}^3 - D_{c,i}^3) * \rho_{BC-free}}{\sum_i D_{c,i}^3 * \rho_{BC}}$, where $D_p$ and $D_c$ represent the diameter of coated BC particle and BC core, respectively; the $i$ denotes the $i^{th}$ particle in the investigated time window. The determination of $\rho_{BC-free}$ will be discussed in Section 2.2.

A Particle Soot Absorption Photometer (PSAP, Radiance Research) was used to determine 1 Hz aerosol absorption coefficients at wavelengths 470, 530, and 660 nm, which were corrected with the wavelength-averaged Virkkula correction (Virkkula, 2010) and smoothed to 10 s to reduce noise. The uncertainty of absorption coefficients is estimated to be 20 % (Fischer et al., 2010).

A High-Resolution Time-of-Flight Aerosol Mass Spectrometer (HR-ToF-AMS, Aerodyne Research) measured mass concentrations of sulphate ($SO_4^{2-}$), nitrate ($NO_3^-$), ammonium ($NH_4^+$), chloride (Cl$^-$), and OA from particles with vacuum aerodynamic diameters between 50 to 500 nm.

We determine single particle mixing state and elemental composition from offline analysis of particles sampled on Paella TEM grids. Around 50 particles per filter were analyzed. A JEOL™ JEM-2010F FEG-TEM with a ThermoNoran™ EDX detector was used. TEM was performed at 200 KeV accelerating voltage, with a take-off angle of 15.9 degrees for X-ray emission from the sample. The elemental weight percentage per particle was determined using the NSS software (Thermo Electron Corporation) with Cliff-Lorimer Absorbance correction method. Detailed descriptions can be found in Dang et al. (2021).

All in-situ instruments were mounted in the NASA P-3 aircraft and operated under dry conditions. Data for BC mass concentration less than 0.1 μg m$^{-3}$ or absorption coefficient at 660 nm less than 1.5 Mm$^{-1}$ were not included.  Measurements were averaged to 10 s and adjusted to STP values at 273.15 K and 1013 hPa.

**2.2 Optical Models**

Most of the particles were found to be nearly spherical from the images obtained by TEM, with >70 % of the particles having aspect ratios (the largest and smallest characteristic sizes of arbitrarily shaped particle) smaller than 1.5. Therefore, we applied Mie theory to determine aerosol optical properties. Models such as T-matrix or discrete dipole approximation model were not used as they would require a large number of free parameters (Scarnato et al., 2013; He et al., 2016). We investigated the sensitivity of our closure simulations to four different models for the BC-containing particles – the ideal core-shell (CS) model and three homogeneous mixing models: 1) the volume mixing (VM) model, 2) the Maxwell-Garnett (MG) model, and 3) the Bruggeman (BG) model. The number size distribution of BC-containing particles was obtained from the BC core size distribution and the BC 2-D size and mixing state (i.e. coating thickness) distribution from SP2. Detailed descriptions and inputs of the four models can be found in Section S1 in the supplement. We assumed that the non-BC components were homogeneously mixed and calculated the refractive index of the mixture, $m_{\text{BC-free}}$, with the VM rule. The mass concentrations of $SO_4^{2-}$, $NO_3^-$, $NH_4^+$, and Cl$^-$ measured by AMS were converted to those of inorganic salts to facilitate calculation, using the simplified ion-pairing scheme proposed by Gysel et al. (2007), modified as follows:

$$n_{NH_4Cl} = n_{Cl^-} \tag{1}$$
$$n_{NH_4NO_3} = \min\left(n_{NO_3^-}, n_{NH_4^+} - n_{Cl^-}\right) \tag{2}$$
$$n_{NH_4HSO_4} = \max\left(0, \min\left(2n_{SO_4^{2-}} - n_{NH_4^+} + n_{NO_3^-} + n_{Cl^-}, n_{NH_4^+} - n_{NO_3^-} - n_{Cl^-}\right)\right) \tag{3}$$
$$n_{(NH_4)_2SO_4} = \max\left(0, \min\left(n_{SO_4^{2-}}, n_{NH_4^+} - n_{NO_3^-} - n_{Cl^-} - n_{SO_4^{2-}}\right)\right) \tag{4}$$
$$n_{KNO_3} = \max\left(0, \min\left(n_{NO_3^-}, n_{NH_4^+} - n_{Cl^-} - n_{NO_3^-} - 2n_{SO_4^{2-}}\right)\right) \tag{5}$$
$$n_{K2SO_4} = \max\left(0, \min\left(n_{SO_4^{2-}}, n_{SO_4^{2-}} - n_{NH_4^+} + n_{NO_3^-} + n_{Cl^-}\right)\right) \tag{6}$$

where $n$ represents the number of moles. As potassium salts are the most frequently detected salts from TEM-EDX analysis (Dang et al., 2021), the residual anions were assumed to be combined with potassium, i.e. KNO$_3$ and K$_2$SO$_4$; however, a small fraction of $NO_3^-$ and $SO_4^{2-}$ detected by the AMS might be organics and thus lack an accompanying cation. Densities of the various salts and OA used to convert the mass concentrations to volume concentrations were taken from Kuang et al. (2020; Table 1) and Liu et al. (2015). The $m_{\text{BC-free}}$ is calculated as:

$$m_{BC-free} = \left(\sum_{i=salts} m_i * V_i + m_{OA} * V_{OA}\right) \Big/ \left(\sum_{i=salts} V_i + V_{OA}\right) \tag{7}$$

where $m_i$ and $V_i$ represent the refractive indices and volumes of aforementioned salts; $m_{OA}$ and $V_{OA}$ are those for OA. The refractive indices of salts are taken from Fierz-Schmidhauser et al. (2010) and Cotterell et al. (2017). The real part of the refractive index of OA varies from 1.35 to 1.7 (Lack and Cappa, 2010; Liu et al., 2013; Saleh et al., 2014; Moise et al., 2015); 1.55 is used in this study. OA is assumed to be non-absorbing in the calculation. The average and standard deviation of calculated $m_{\text{BC-free}}$ is 1.52±0.015, consistent with those used in BrC studies in literature (Saleh et al., 2013; Liu et al., 2021, 2015). The particle effective refractive indices for the VM, MG, and BG models can then be obtained using the corresponding mixing rules (Section S1 in the supplement) with $m_{\text{BC-free}}$ and $m_{\text{BC}}$. The determination of the refractive index of BC will be discussed in the following section.

## 2.3 Optical calculation procedure

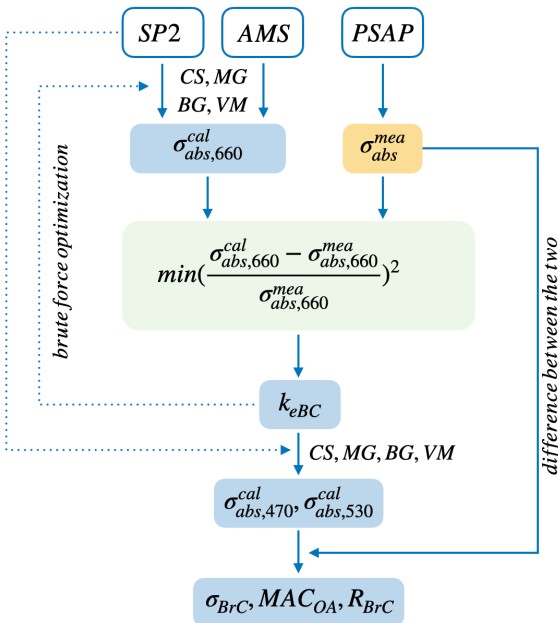

**Figure 2. Schematic diagram of the iterative derivation of BrC absorption attribution performed in this study. Cells in white, yellow, and blue represent instrument, measurements, and calculations, respectively. The green box is the optimization step used to determine the values of the imaginary part of the effective refractive index of BC, $k_{eBC}$, once the deviation of measured and calculated absorption coefficients at 660 nm ($\sigma^{cal}_{abs,660}$ and $\sigma^{mea}_{abs,660}$) are minimized. $k_{eBC}$ is held constant at investigated wavelengths, i.e. 470, 530, and 660 nm.**

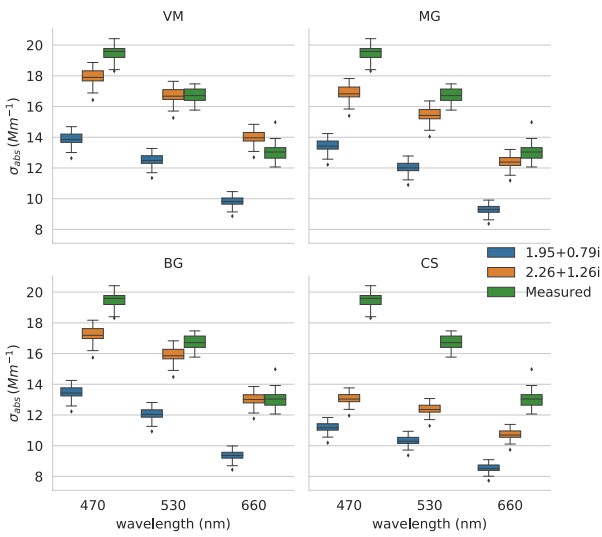

**Figure 3. Modelled (blue and orange markers) and measured (green markers) absorption coefficients ($\sigma_{abs}$) at PSAP wavelengths for RF06_1. Variables are modelled with two $m_{BC}$ values (shown in the legend) using the CS, MG, BG,**

**and VM models (specified on the top of each plot). OA is assumed to be non-absorbing with the refractive index $m_{OA}$ of 1.55+0i. The horizontal lines in the boxes represent the median value, the boxes represent 25th to 75th percentile, the whiskers represent 1.5 inter-quartile range, and the diamonds represent outliers.**

The numerical value selected for $m_{BC}$ has a large impact on the absorption attributed to BC (Fig. 3), and hence on the estimated absorption by BrC, but as noted above, there is no consensus on the best value of $m_{BC}$. Additionally, the lack of measurements of possible absorbers other than BC and BrC such as magnetite makes it difficult to correctly estimate the BrC absorption, so approaches that take these components into consideration are required. Here, we introduce an effective refractive index of BC, $m_{eBC} = 1.95+ik_{eBC}$, to represent the refractive index of BC and any other absorbing components at 660 nm in this study. The $m_{eBC}$ would be the same as $m_{BC}$ if the absorption at 660 nm is solely contributed by BC; however, if other absorbing components are present, the imaginary part ($k_{eBC}$) of $m_{eBC}$ would be greater than that ($k_{BC}$) of $m_{BC}$, the amount of which depends on the absorptivity and relative amount of these absorbers. The real part of $m_{eBC}$ is held constant at 1.95, the upper bound of the values commonly used for BC (Bond and Bergstrom, 2006; Saleh et al., 2013; Liu et al., 2015; Kahnert and Kanngießer, 2020). It has only a minor influence on absorption calculations (Liu et al., 2021): sensitivity test of the calculated absorption to the real part of $m_{eBC}$ when the latter is varied from 1.75 to 2.26 shows that, at the extreme case of RF10 with the thickest coating in this study, the absorption increased less than 5 % at 660 nm.

The retrieval of $k_{eBC}$ is illustrated in Fig. 2. OA is assumed to be non-absorbing; thus, the imaginary refractive index $k_{OA}$ is taken as zero in this calculation. Values of $k_{eBC}$ were determined by minimizing the squares, $\chi^2$, of the relative differences between the measured and calculated absorption coefficients at 660 nm for each model:

$$\chi^2(k_{eBC}) = (\frac{\sigma_{abs,660}^{cal} - \sigma_{abs,660}^{mea}}{\sigma_{abs,660}^{mea}})^2, \qquad (8)$$

where $\sigma_{abs,660}^{cal}$ and $\sigma_{abs,660}^{mea}$ are the calculated and measured absorption coefficients at 660 nm, respectively. The value of $k_{eBC}$ is searched from 0.3 to 2.6, over a greater range than the values of $k_{BC}$ recorded in literature (Chylek et al., 2019; Kahnert and Kanngießer, 2020), and with a span of 0.01. The $k_{eBC}$ values with $\chi^2$ greater than 0.1 are excluded.

The absorption of BrC at the long visible, i.e. 660 nm, is expected to be very small or negligible considering the aged nature of aerosols in our study (Sedlacek III et al., 2018; Adler et al., 2019; Li et al., 2019). Therefore, we assume that the contribution of BrC to $k_{eBC}$ can be neglected, and use it to differentiate the BrC absorption from that of other substances, i.e. BC and any other absorbers. This assumption of nil contribution of BrC to $k_{eBC}$ yields a lower bound of BrC absorption at 530 and 470 nm. An evaluation of the influence of this assumption to the BrC absorption at shorter wavelengths is discussed in Section S2 in the supplement. As illustrated in Fig. 2, $m_{eBC}$ was held constant over the range of investigated wavelengths, i.e. 470-660 nm. This is

supported by $m_{BC}$ that it is usually considered to be very weakly dependent on, or independent of, wavelength in the visible spectrum (e.g., Chang and Charalampopoulos, 1990; Moteki et al., 2010; Saleh et al., 2014). As discussed later in Section 3.1, other absorbers contributing to $m_{eBC}$, if any, are hypothesized to be magnetite, whose refractive index is also mostly invariant between wavelengths 470 and 660 nm (Amaury et al., unpublished data, http://www.astro.uni-jena.de/Laboratory/OCDB/mgfeoxides.html; Ackerman and Toon, 1981; Zhang et al., 2015). Thus, the assumption of constant $m_{eBC}$ between 470 and 660 nm is reasonable in our study, although it may lead to overestimations of BrC absorption for highly aged particles, as discussed further in Section 3.2.

The absorption of BrC at wavelength λ ($\sigma_{abs,BrC,\lambda}$) is calculated as the difference between measured absorption coefficient ($\sigma_{abs,\lambda}^{mea}$) and that calculated with $m_{eBC}$ ($\sigma_{abs,\lambda}^{cal}$):

$$\sigma_{abs,BrC,\lambda} = \sigma_{abs,\lambda}^{mea} - \sigma_{abs,\lambda}^{cal} \tag{9}$$

The fractional contribution of BrC to the total absorption at wavelength λ, $R_{BrC,\lambda}$, is defined as

$$R_{BrC,\lambda} = \sigma_{abs,BrC,\lambda}/\sigma_{abs,\lambda}^{mea} \tag{10}$$

and is assumed to be zero at 660 nm in our study. The mass absorption cross section (MAC) of OA at wavelength λ, $MAC_{OA,\lambda}$, can be determined from $\sigma_{abs,BrC,\lambda}$ and the mass concentration of OA, $M_{OA}$:

$$MAC_{OA,\lambda} = \sigma_{abs,BrC,\lambda}/M_{OA} \tag{11}$$

Similarly, the mass absorption coefficient of BC ($MAC_{BC}$) is defined as: $MAC_{BC,\lambda} = \sigma_{abs,\lambda}^{mea}/M_{BC}$, where $\sigma_{abs,\lambda}^{mea}$ and $M_{BC}$ represent the measured absorption coefficient at wavelength λ and the mass concentration of BC, respectively.

## 3 Results and discussion

### 3.1 Effective refractive index of BC ($m_{eBC}$)

In this section, we present and discuss the results of $k_{eBC}$ derived from the aforementioned four models for each flight. Uncertainties of $k_{eBC}$ retrieved from various models have been determined by Monte-Carlo uncertainty analysis in Section S3 in the supplement. The values of $k_{eBC}$ for each flight determined by the various models are shown in Figure 4. No $m_{eBC}$ for RF05_3 was obtained with the CS model because the criteria of $\chi^2$ smaller than 0.1 was not met, as the absorption coefficient calculated with the CS model using the largest possible $m_{BC}$ (i.e. 2.26+1.26i) was only half of the measurement for RF05_3 (Fig. S2). Variations of $k_{eBC}$ show similar patterns among different models (Fig. 4). The values derived from the CS and VM models are the highest and lowest, respectively, and the values derived from the MG and BG models are between the other two models and very close to each other, in agreement with the absorption results calculated with various models in previous studies (Lesins et al., 2002; Fierce et al., 2017a; Taylor et al., 2020). The values of $k_{eBC}$ for RF05_1, RF05_2, RF06_1, and RF06_2 from the CS model have greatly exceeded the largest value of $k_{BC}$ of 1.26 (grey shaded region in Fig. 4), determined by Moteki et al. (2010) for soot particles from an urban source (Tokyo). Little confidence has been placed in the value of $k_{eBC}$ from the CS model except for RF10 for two main reasons. First,

the MR for all samples is less than 2, and application of the CS model can be inappropriate for particles with such low MR values (Liu et al., 2017). Furthermore, the TEM analysis found few core-shell structures (e.g., example of RF05_1 in Fig. 5) among investigated samples, with most BC-containing particles composed of condensed BC nodules internally mixed with salts/organics (e.g., example of RF05_3 in Fig. 5) or aggregates attached to non-

BC components (e.g., example of RF05_2 in Fig. 5). Representative TEM images of different types of BC-containing particles for each flight are shown in Fig. 5. The use of CS model for RF10 was supported by its high MR value of 7.4, which has exceeded the lower bound of MR for the use of CS model proposed by Liu et al. (2017), although a fair number of BC particles homogeneously mixed with salts and OA were detected as well (Fig. 5). In addition, results of the BrC contribution to the total absorption suggest that homogeneous models

might be inappropriate for RF10, as discussed in detail in Section 3.3.

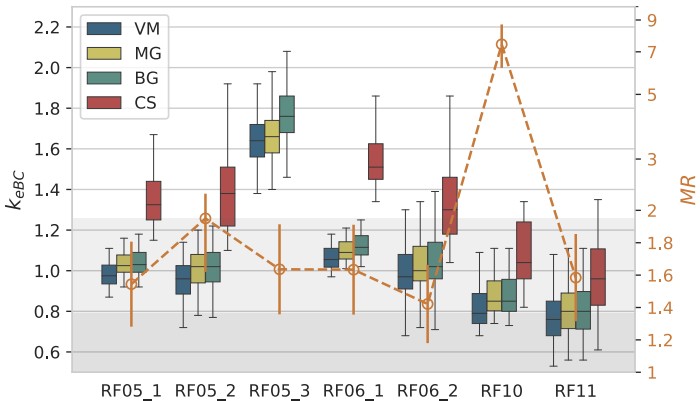

**Figure 4. Boxplot of $k_{eBC}$ (left axis) derived from different models and the MR values (orange open dots, right axis) for each flight. Error bars of MR represent 20 % uncertainty. The right y-axis uses the log scale. Light and dark grey shaded region shows $k_{eBC}$ smaller than 1.26, the largest $k_{BC}$ value determined by Moteki et al. (2010), and 0.79, the**

**largest value of commonly used $k_{BC}$, respectively.**

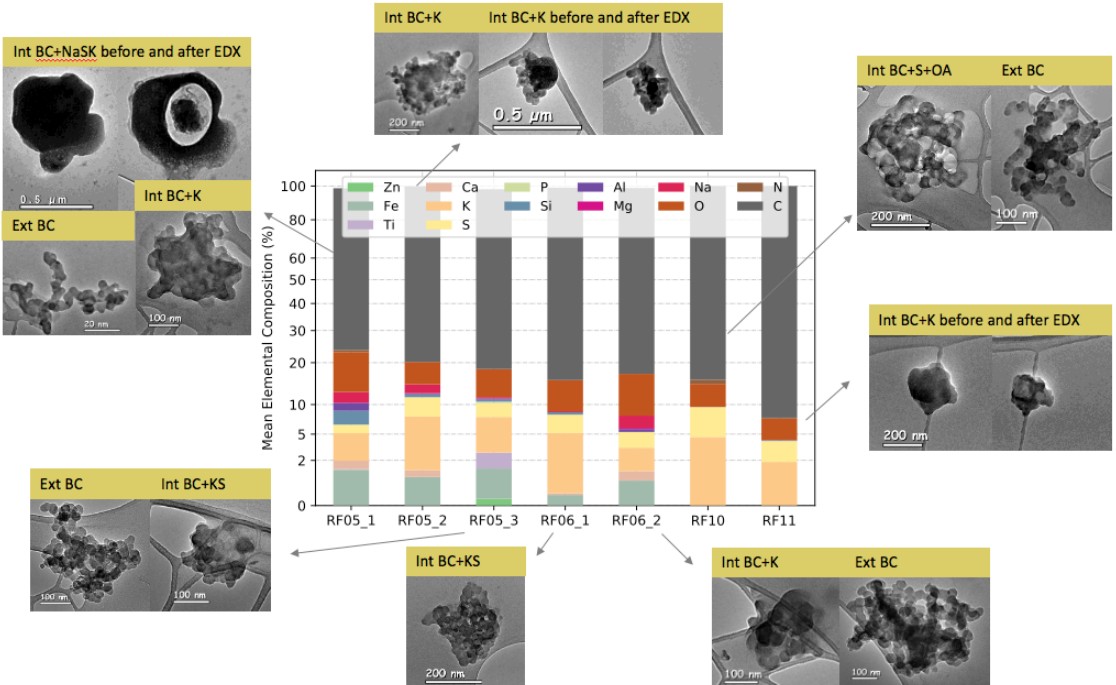

**Figure 5. Elemental composition and representative microphysical images of different types of BC-containing particles from TEM analysis for each flight. Int BC and Ext BC represent internally and externally mixed BC particles, respectively. For example, Int BC+K stands for internally mixed BC with potassium salts. The y-axis uses a $x^{1/2}$ scale.**

As discussed in Section 2.3, $k_{eBC}$ should be the same as $k_{BC}$ if the absorption at 660 nm arises only from BC; higher $k_{eBC}$ value indicates larger absorptivity of absorbers at 660 nm. The values of $k_{eBC}$ for RF05_1, RF05_2, RF05_3, RF06_1, and RF06_2, for which iron (Fe) was detected from the TEM-EDX (as shown in the bar chart in Fig. 5), are mostly larger than unity and greater than those for RF10 and RF11, for which no iron was detected. Therefore, we suspect higher $k_{eBC}$ values may be related to iron oxides (FeOx) that absorbs at 660 nm. We have examined all possible compounds of detected elements from EDX and found that magnetite is the only compound that is not rare in the atmosphere and also absorbs strongly at 660 nm. Magnetite shows a strong and uniform absorptivity over the visible spectrum with reported imaginary refractive index ranging from 0.58 to 1.0 (Amaury et al., unpublished data, http://www.astro.uni-jena.de/Laboratory/OCDB/mgfeoxides.html; Ackerman and Toon, 1981; Zhang et al., 2015), and therefore can contribute to the high values of $k_{eBC}$ at 660 nm. Studies show that magnetite can be transformed from Fe(III) at high temperatures, such as goethite and hematite (Till et al., 2015; Ito et al., 2018), which happen to be the two most abundant forms of FeOx in African dust (Formenti et al., 2014). Therefore, we hypothesize that a part of the magnetite might be converted from Fe(III) during biomass burning. Additionally, magnetite can be emitted from anthropogenic activities, such as steel manufacturing, oil combustion, and engines and brakes of motor vehicles (Machemer, 2004; Liati et al., 2015; Moteki et al., 2017; Kurisu et al., 2019). The significance of anthropogenic magnetite in radiative forcing has been investigated and highlighted in recent studies (Moteki et al., 2017; Ito et al., 2018; Lamb et al., 2021). Although there is no investigation on

magnetite in sub-Saharan Africa yet, the industrial and motor vehicle emissions in Africa are likely to contribute magnetite. Furthermore, the pyrometallurgical process is a widely used extraction method in copper mining, a major industry in the central African Copperbelt (https://www.pyrometallurgy.co.za/PyroSA/, Vítková et al., 2010; Sikamo et al., 2016; Shengo et al., 2019), of which iron is a common unwanted slag element (Meter et al., 1999).

Therefore, we suspect it might also contribute magnetite considering the high temperature of the pyrometallurgical process.

RF05_3, collected at the uppermost aerosol layer (Fig. 4), is an exception among all flights, since the values of the absorption coefficient at 660 nm calculated with all four models are considerably smaller than the measured ones (Fig. S2). The MG, BG, and VM models yield median $k_{eBC}$ values greater than 1.6; no values met

the criteria of $\chi^2<0.1$ for the CS model. The $MAC_{BC}$ of RF05_3 is the highest among all flights with values (mean±standard deviation) of 20.0±0.8, 17.8±0.8, and 14.3±0.7 $m^2$ $g^{-1}$ at 470, 530, and 660 nm, respectively. The absorption enhancement, $E_{Abs}$, defined as the ratio of $MAC_{BC}$ to the value for uncoated BC reported by Bond and Bergstrom (2006), is 2.3±0.1 for all three wavelengths. To the best of our knowledge, except for modelling or laboratory studies of thickly coated particles (Bond et al., 2006; Jacobson, 2012; Peng et al., 2016), such high

values of $E_{Abs}$, particularly at long visible wavelengths, are rarely reported in field measurements (Cui et al., 2016). Taylor et al. (2020) presented relatively high $MAC_{BC}$ values of 20±4, 15±3, and 12±2 $m^2$ $g^{-1}$ at 405, 514, and 655 nm, respectively and an $E_{Abs}$ of 1.85±0.45 for CLARIFY 2017 (Cloud-Aerosol-Radiation Interaction and Forcing 2017 measurement campaign; Haywood et al., 2020), which are generally comparable or smaller than those for RF05_3. However, particles in CLARIFY 2017 are universally thickly coated, with median MR values of 8-12,

and are therefore would be expected to have large values of $E_{abs}$, as opposed to RF05_3, which had a median MR (Fig. 4) of 1.6. We calculated absorption coefficients for RF05_3 at 660 nm using the VM model, which usually yields a higher estimation of the measurements (Taylor et al., 2020), and a $m_{BC}$ of 1.95+0.79i, an upper bound of the commonly used BC refractive index (Liu et al., 2021). Uncertainties related to calculated absorption coefficients can be found in Section S3 in the supplement. The calculated absorption, which is very likely an

overestimation of the measurement, is only able to explain ~57 % of the measured absorption coefficient (Fig. S2), implying that the remaining amount of absorption which cannot be attributed to BC might result from other absorbers. This unexplained portion of absorption in RF05_3 is at least twice as high as that in other flights (e.g. Fig. 3 and Fig. S2). Back-trajectories show that air parcels in RF05_3 passed by the Copperbelt in Zambia (Fig. 1), where the pyrometallurgical process in copper mining might contribute large amounts of magnetite. In addition,

we noticed titanium (Ti) on the particles from RF05_3. Although titanium oxides are not absorbing, several forms of Ti have been reported to strongly absorb at visible wavelengths (Pflüger and Fink, 1997; Palm et al., 2018). Therefore, we suspect this unexplained absorption in RF05_3 may also be related to absorbing titanium compounds.

**3.2 Mass absorption coefficient of OA (MAC$_{OA}$)**

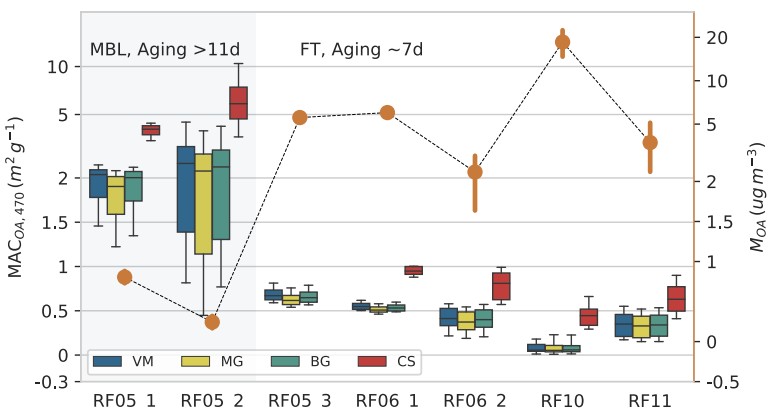

**Figure 6. Values of MAC$_{OA,470}$ (boxes) derived using different models and the mass concentration of OA, $M_{OA}$ (orange dots, left y-axis), for each flight. Error bars represent 1σ standard deviation. The scale of both x-axes is symmetric log. The horizontal lines of boxplot represent the median value, the boxes represent 25$^{th}$ to 75$^{th}$ percentile, the whiskers represent 1.5 inter-quartile range. RF05_1 and RF05_2 were measured in the MBL with plume agd larger than 11 days, other flights were measured in the FT with aging time around 7 days. The aerosol/plume age provided by a two-week forecast using the Weather Research and Aerosol Aware Microphysics (WRF-AAM) model.**

Values of MAC$_{OA,470}$ calculated from Eq. 11 using the different models are shown in Fig. 6. The uncertainty of MAC$_{OA,470}$ has been determined using Monte-Carlo uncertainty analysis in Section S3 in the supplement. The values of MAC$_{OA,470}$ from the three homogeneous models were fairly close to each other, as expected from previous studies (Lesins et al., 2002); while MAC$_{OA,470}$ from the CS model was much greater, highlighting the importance of model treatment. From Section 3.1, results of RF10 from the CS model and the results of other flights calculated with homogeneous models are more plausible, with the values (mean±standard deviation) of MAC$_{OA,470}$ for RF05_3, RF06_1, RF06_2, RF10, and RF11 ranging from 0.30±0.27 m$^2$ g$^{-1}$ to 0.68±0.08 m$^2$ g$^{-1}$. This result at 470 nm is generally comparable to the value of 0.31±0.09 m$^2$ g$^{-1}$ at 405 nm for highly aged aerosols sampled downwind of ORACLES in CLARIFY 2017 campaign (Taylor et al., 2020).

RF05_1 and RF05_2, whose particles are as least as aged as those in CLARIFY, showed unexpectedly high MAC$_{OA,470}$ values of 1.84±0.64 m$^2$ g$^{-1}$ and 2.38±1.89 m$^2$ g$^{-1}$, respectively. Although the values of $M_{OA}$ for these samples are low (Fig. 6), measurements from all instruments are within the detection limit, which would argue that these values are valid. Wang et al. (2018) used airborne measurements to constrain their global model and found the best MAC$_{OA}$ to represent the measurements was 1.33 m$^2$ g$^{-1}$ for freshly emitted BB OA at 365 nm. Lin et al. (2017) investigated relatively fresh BB aerosols subject to atmospheric processing during a night-long BB event in an urban environment and reported a MAC of 0.9 m$^2$ g$^{-1}$ for water extractable BrC at 470 nm under

the peak BB episode. Our values of MAC$_{OA,470}$ for RF05_1 and RF05_2 are much higher than those for much fresher BB aerosols, which seems unrealistic, as BrC bleaching is expected to occur during transport (Hems et al., 2021; Che et al., 2021). One possible explanation is that secondary BrC formation occurred, perhaps through aqueous-phase chemistry during transport (Hems et al., 2021), as RF05_1 and RF05_2 were sampled in the MBL

with higher RHs. Saleh et al. (2013) reported that secondary BrC can be more absorbing than primary BrC at short visible wavelengths; however, to the best of our knowledge, such high MAC like those in RF05_1 and RF05_2 for secondary BrC have not been documented (Kasthuriarachchi et al., 2020).

We assume a constant $m_{eBC}$ over the investigated spectrum, i.e. 470-660 nm; while the $m_{eBC}$ would be underestimated at shorter wavelengths if there are some components with strong absorption at short visible but

not at long visible wavelengths, and therefore lead to an overestimation of MAC$_{OA.470}$. Hematite, whose imaginary refractive index ranges from 0 to 1.0 at 470 nm and 0 to ~0.2 at 670 nm (Zhang et al., 2015b; Go et al., 2021), is the second most abundant FeOx in western African dust (Formenti et al., 2014). Particles in RF05_1 and RF05_2 experienced over 11 days of transport and thus may have more opportunities to mix with hematite and therefore lead to an overestimation of MAC$_{OA}$. While lacking measurements to determine the compound, it is difficult to

verify our speculation. A modified SP2 has been reported by Yoshida et al. (2016) that can discriminate black-coloured magnetite and red-coloured hematite; thus, we recommend adding such measurement capabilities in future BB investigations over Africa and the South-East Atlantic region.

### 3.3 Contribution of BrC to total absorption ($R_{BrC}$)

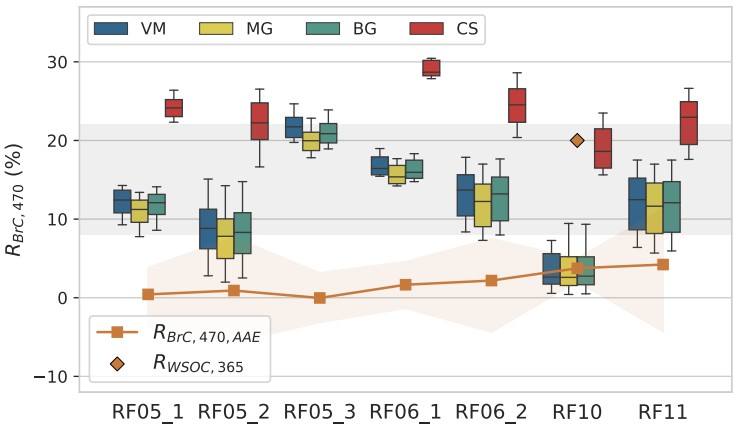

**Figure 7. Contribution of BrC to the total absorption obtained from optical closure approach in this study (boxes, $R_{BrC,470}$) and from the AAE attribution method (orange squares, $R_{BrC,470,AAE}$) at 470 nm. The grey zone indicates the $R_{BrC,470}$ range of 8-22 % in this study. Orange shades of $R_{BrC,470,AAE}$ represent the 10th and 90th percentile. The orange diamond is the contribution of WSOC (water soluble organic carbon) to the total absorption, $R_{WSOC,365}$ (Nenes A.,**

**personal communication).**

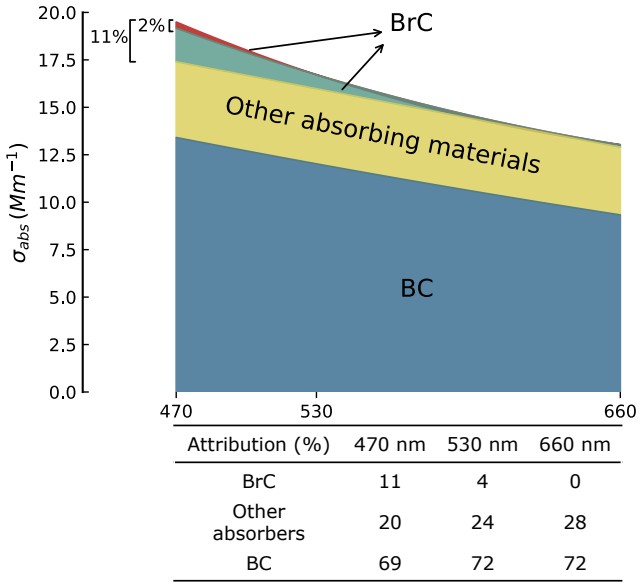

**Figure 8. Illustration of the attribution of absorption for RF06_1. Blue area represents the absorption of BC, calculated with $m_{BC}$=1.95+0.79i using BG model. The upper border of the yellow region represents absorption coefficients calculated with corresponding $m_{eBC}$ for RF06_1 using BG model assuming non-absorbing OA; the yellow area illustrates the absorption by absorbing components besides BC and BrC, calculated as the difference between the two absorption coefficients calculated by the BG model using $m_{eBC}$ and $m_{BC}$, respectively. The upper border of the red region is drawn with measured absorption coefficients at 470, 530, and 660 nm. The red region stands for the BrC absorption coefficient calculated from the AAE attribution method with Eq. 13. The 2 % and 11 % are the proportions of BrC at 470 nm estimated from the AAE attribution method and our optical closure method, respectively. The table in the lower panel shows the absorption attribution at 470, 530, and 660 nm from the optical closure method. Uncertainties can be found in Table S2 in the supplement.**

The contribution of BrC to the total absorption at 470 nm, $R_{BrC,470}$, which was calculated using the optical closure method from Eq. 10, is shown in Fig. 7. The uncertainties using various models can be found in Section S3 in the supplement. Similar to the results for MAC$_{OA,470}$, the MG, BG, and VM models yield fairly close results for $R_{BrC,470}$, while those from the CS model are higher, consistent with previous studies (Lesins et al., 2002; Taylor et al., 2020). It is noted that the values of $R_{BrC,470}$ from MG, BG, and VM models at 470 nm for RF10 are the lowest among all flights, with an average of 3.6±2.6 %. Little confidence is placed on this value because particles with similar age in RF05_3, RF06_1, RF06_2, and RF11 all showed a much higher value of $R_{BrC,470}$. In addition, Nenes et al. (personal communication) reported the contribution of water-soluble organic carbon (WSOC) to the total absorption at 365 nm of 20 % (Fig. 7) for particles measured at a similar location in the plume 10 min later than RF10, with no distinct changes in either the meteorological conditions or aerosol properties. Therefore, we expect the AAE of BrC at 365/470 wavelength pair, AAE$_{365/470,BrC}$, derived from $\sigma_{abs,365,BrC}^{mea}$ and $\sigma_{abs,470,BrC}^{cal}$

(absorption coefficients of BrC at 365 and 470 nm) to be within the range of ~2-11 in literature (Laskin et al., 2015). Since $\sigma_{abs,365,BrC}^{mea}$ is unknown, we calculated $\sigma_{abs,365,extrapolate}$ by extrapolating the measured $\sigma_{abs,470}^{mea}$ to 365 nm with the measured AAE$_{470/530}$ and approximated the AAE$_{365/470,BrC}$ with the 20 % of $\sigma_{abs,365,extrapolate}$ and the $\sigma_{abs,470,BrC}^{cal}$ from homogeneous models. This method yields an underestimation of AAE$_{365/470,BrC}$; while its value can still reach 18, much higher than the upper limit of the AAE range reported in literature (Laskin et al., 2015), suggesting that the use of homogeneous models for RF10 may not be appropriate. As discussed in Section 3.1, the values of $R_{BrC,470}$ for RF10 from the CS model and RF05_1, RF05_2, RF05_3, RF06_1, RF06_2, and RF11 from homogeneous models are more plausible. Particles of RF05_1 and RF05_2 may contain hematite, in which cases their R$_{BrC,470}$ may be overestimated (Section 3.2). Generally, in our study, the $R_{BrC,470}$ ranges from ~8-22 % at 470 nm (grey zone in Fig. 7), slightly lower than the contribution of 13-26 % at 500 nm for relatively fresh OA in the wood smoke from Kirchstetter and Thatcher (2012) and generally comparable or slightly higher than the result of ~11 % at 405 nm for highly aged aerosols from CLARIFY 2017, which were sampled downwind of ORACLES, implying the bleaching of BrC during transport (Hems et al., 2021; Che et al., 2021).

We calculated the contribution of BrC to the total absorption using the AAE attribution method ($R_{BrC,470,AAE}$) and compared it with that from our optical closure method ($R_{BrC,470}$). In the AAE attribution method, the absorption coefficient of BC at the investigated wavelength (470 nm in our case) is determined by extrapolating the absorption coefficient at a longer wavelength (530 nm in our case) with the AAE of BC. For example, in this study, the average AAE at 530/660 wavelength pair (AAE$_{530/660}$) for all flights is 0.93±0.16, which is just below unity and within the range expected for BC (Liu et al., 2018; Taylor et al., 2020). Thus, we assume that the absorption coefficients at 530 nm ($\sigma_{abs,530}^{mea}$) and AAE$_{530/660}$ have no significant contribution from BrC and that the AAE for BC is independent of wavelength. The contribution of BrC to the total absorption, $R_{BrC,470,AAE}$, can then be calculated in below.

$$R_{BrC,470,AAE} = 1 - \frac{\sigma_{abs,470}^{cal,AAE}}{\sigma_{abs,470}^{mea}} \tag{12}$$

$$\sigma_{abs,470}^{cal,AAE} = \sigma_{abs,530}^{mea}\left(\frac{530}{470}\right)^{AAE_{530/660}} \tag{13}$$

$$AAE_{530/660} = -\frac{\ln(\sigma_{abs,530}^{mea}/\sigma_{abs,660}^{mea})}{\ln(530/660)} \tag{14}$$

$\sigma_{abs,470}^{mea}$, $\sigma_{abs,530}^{mea}$, and $\sigma_{abs,660}^{mea}$ represent the measured absorption coefficients at 470, 530, and 660 nm, respectively. $\sigma_{abs,470}^{cal,AAE}$ is the absorption coefficient calculated from the AAE attribution method at 470 nm. Actually, this assumption is not always true as the AAE varies with both the wavelength and core size and coating thickness of BC particles (Lack and Cappa, 2010; Wang et al., 2016). The results of $R_{BrC,470,AAE}$ are shown in Fig. 7, with averages of all flights below 5 %, roughly a factor of three smaller than the $R_{BrC,470}$ from the optical closure approach.

The AAE attribution method is widely used for its simplicity, while there is one caveat that need to be noted when applying this method. The absorption coefficients and AAE, in this case, $\sigma_{abs,530}^{mea}$, $\sigma_{abs,660}^{mea}$, and AAE$_{530/660}$, should be for BC only, without contributions from other absorbers, such as BrC (Lack and Cappa, 2010). However, it is usually very difficult to completely exclude the impact of these absorbers in the calculation. The homogeneous models are usually reported to overestimate BC absorption (Zhang et al., 2015a; Fierce et al., 2017b; Taylor et al., 2020) and hence underestimate BrC absorption; while the values of the contribution of BrC determined by the AAE attribution method are even lower than those from the homogeneous methods, which we suspect might be partly attributed to the impact of absorbers besides BC. Therefore, we evaluated the absorption attribution at detected wavelengths by taking RF06_1 as an example. The contribution of BrC derived from the AAE attribution method is 2 % for RF06_1, approximately a factor of five smaller than that from the optical closure method with the BG model, 11 % (Fig. 7 and 8). As illustrated in Fig. 8, $\sigma_{abs,530}^{mea}$ includes the contribution of BrC, which accounts for 4 % of the total absorption, and that of absorbers beyond BC and BrC, which accounts for 21 %. Note this attribution is based on the BC absorption coefficient calculated from the BG model with $m_{BC}$=1.95+0.79i, which yields an upper bound of BC absorption (Liu et al., 2021). All calculated parameters are subject to uncertainties as discussed in Section S3 in the supplement. If we remove the 4 % BrC from $\sigma_{abs,530}^{mea}$ in Eq. 13, the BrC contribution to absorption at 470 nm would increase to 6 %, two times larger than 2 %, indicating a substantial impact of BrC on the result of the AAE attribution method even though the BrC absorption may only exist as a small portion. Thus, we recommend that application of any optical properties-based attribution method to use absorption coefficients at the longest possible wavelength to minimize the influence of BrC, and in the meanwhile, to account for potential contributions from other absorbing materials.

## 4 Conclusions

We investigated the contribution of BrC to the total absorption of BB aerosols with different models utilizing measurements from ORACLES 2018 field campaign. An effective BC refractive index, $m_{eBC}$=1.95+i$k_{eBC}$, that was constrained by absorption measurements to account for all absorbing components at 660 nm, was introduced in this study. Most of the values derived for $k_{eBC}$ were greater than the commonly used $k_{BC}$ values, suggesting contributions from absorbing materials besides BC at 660 nm. TEM-EDX single particle analysis further suggests that these absorbers might include FeOx that absorbs at long visible wavelengths, i.e. magnetite, as Fe is only present for flights with large values of $k_{eBC}$. RF05_3 yielded the largest values of $k_{eBC}$, approximately 60-100 % greater than that of other flights, implying a greater contribution of absorbers besides BC to the total absorption at 660 nm than other flights, which might be due to a larger amount of magnetite in these samples or the result of possible absorbing titanium compounds, which are present on this filter.

As refractive indices of BC and magnetite are generally constant over the range of wavelengths considered, i.e. 470 and 660 nm, we assumed $m_{eBC}$ to be independent of wavelength and calculated $MAC_{OA}$ and the contribution of BrC to total absorption at 470 nm, $R_{BrC,470}$, with different models. The values of $MAC_{OA,470}$ and $R_{BrC,470}$ from the three homogeneous mixing models were fairly close to each other, while those from the CS model were much higher, underscoring the importance of model treatment. The values of $MAC_{OA,470}$ and $R_{BrC,470}$ from the CS model for RF10 and from homogeneous models for other flights are found to be more plausible, based on the morphology of BC-containing particles from TEM analysis, MR values from SP2, and the contribution of WSOC to the total absorption at 365 nm. The values of $R_{BrC,470}$ ranged from ~8-22 %, with $MAC_{OA,470}$ varying between $0.30\pm0.27$ $m^2$ $g^{-1}$ and $0.68\pm0.08$ $m^2$ $g^{-1}$ for the various flights. Values from our results were generally comparable or slightly higher than the $MAC_{OA,470}$ of $0.31\pm0.09$ $m^2$ $g^{-1}$ and $R_{BrC}$ of ~11 % at 405 nm for highly aged aerosols from CLARIFY 2017, which were sampled downwind of ORACLES, while under a much longer wavelength, i.e. 470 nm, implying the bleaching of BrC during transport. High $MAC_{OA,470}$ values of $1.84\pm0.64$ and $2.38\pm1.89$ $m^2$ $g^{-1}$ were observed for RF05_1 and RF05_2, respectively, which is suspected to be caused by hematite, an abundant FeOx in African dust, which may have mixed with the BC particles in RF05_1 and RF05_2 during the 11 d transport. Measurements of a modified SP2 that can distinguish between hematite and magnetite and obtain FeOx concentrations are highly recommended.

Contribution of BrC to the total absorption obtained from the AAE attribution method, $R_{BrC,470,AAE}$, were calculated and compared to that from our optical closure method. $R_{BrC,470,AAE}$ is generally <5 %, approximately a factor of three smaller than those from our optical closure method. Absorption attribution using the Bruggeman mixing Mie model suggest a minor BrC contribution of 4 % at 530 nm for RF06_1, and its removal would triple the BrC contribution to the total absorption at 470 nm from 2 % to 6 % obtained using the AAE attribution method, suggesting a substantial impact of BrC to the result of AAE attribution method even though the BrC may only exist as a small portion. Thus, when applying optical properties-based attribution methods, it is recommended to use the absorption coefficient at the longest possible wavelength to minimise the influence of BrC and to account for other absorbers.

*Acknowledgements.* Lu Zhang would like to thank the postdoctoral fellowship funding from Tel Aviv University, Department of Exact Sciences. Michal Segal-Rozenhaimer and Haochi Che were funded by United States Department of Energy (DOE) Atmospheric System Research (ASR) grant DE-SC0020084. Caroline Dang was funded by a NASA postdoctoral fellowship. Arthur J. Sedlacek III and Ernie R. Lewis were funded by the Scientific Focus Area (SFA) Science Plan program that is supported by the Office of Biological and Environmental Research in the Department of Energy, Office of Science, through the United States Department of Energy Contract No. DE-SC0012704 to Brookhaven National Laboratory. Paola Formenti is supported by the

AErosols, RadiatiOn and CLOuds in southern Africa (AEROCLO-sA) project funded by the French National Research Agency under grant agreement n° ANR-15-CE01-0014-01, the French national programs LEFE/INSU and PNTS, the French National Agency for Space Studies (CNES), the European Union's 7th Framework Programme (FP7/2014-2018) under EUFAR2 contract n°312609, and the South African National Research Foundation (NRF) under grant UID 105958. The authors would like to thank the ORACLES team and the two anonymous reviewers for their helpful comments.

*Data availability.* Airborne measurements are available via the digital object identifier provided under ORACLES Science Team reference: https://doi.org/10.5067/Suborbital/ORACLES/P3/2018_V2.

*Author contributions.* MSR and LZ developed the ideas for the direction of the paper. MSR, AJS, EL, AD, JPW, PF, SGH, and AN carried out the airborne measurements. MSR, HC, CD, AJS, EL, AD, JPW, SGH, and AN processed the aircraft data. LZ and HC performed the data analysis. LZ prepared the paper with inputs and comments from all co-authors.

*Competing interests.* Paola Formenti is guest editor for the ACP Special Issue "New observations and related modelling studies of the aerosol-cloud-climate system in the Southeast Atlantic and southern Africa regions". The remaining authors declare that they have no conflicts of interests.

*Special issue statement.* This article is part of the special issue "New observations and related modeling studies of the aerosol-cloud-climate system in the Southeast Atlantic and southern Africa regions (ACP/AMT inter-journal SI)". It is not associated with a conference.

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
