# Peer review of "Light Absorption by Brown Carbon over the South-East Atlantic Ocean"

_Atmospheric Chemistry and Physics, 2021_

## Referee Comment (RC1)

Review of 'Light Absorption by Brown Carbon over the South-East Atlantic Ocean', by L. Zhang, *et al.*, submitted to *Atmospheric Chemistry and Physics*.

**Summary and general comments**

The authors report spectroscopy measurements of aerosol optical properties from a research aircraft platform for biomass burning aerosols over the South East Atlantic Ocean. A modelling framework is developed for attributing the contribution of light absorbing organics (brown carbon, BrC) to the overall aerosol light absorption. This framework exploits either Core-Shell or 'grey sphere' Mie models, with the latter utilising one of three different effective medium approximations to treat the refractive index. The manuscript is interesting, uses established measurement methods, and adds to the debate surrounding the suitability of Mie models for treating aerosol optical properties. However, the authors assert conclusions that are not reinforced by their analysis, not least because there is scant treatment of systematic uncertainties in their measurements nor appreciation of the limitations imposed by their modelling assumptions. Considerable moderation of some of the claims of key outcomes is needed. Specific Comments are provided below in addition to Technical Corrections that need to be addressed.

**Specific comments** (referring to the page "P" and line "L" numbers)

P1 L32. '$0.25 \pm 0.34$ m$^2$ g$^{-1}$': The error is larger than the mean value. The MAC cannot physically be negative, so this uncertainty is nonsensical. Either this uncertainty needs reconsidering, or some explanation of the significance of a negative MAC is needed.

P5 L11-12. '…except for the Neph, whose humidity is not controlled.' The lack of control of the humidity inside the nephelometer is a big issue in terms of the potential for large systematic uncertainties in measured optical properties that then impact on the later retrievals of $m_{eBC}$ and attribution of absorption to brown carbon. There is no discussion in manuscript of humidity impacts on scattering coefficients and subsequent optical closure calculations. Indeed, no details on the RH in the Neph are provided for the different flights. The Supporting Information just states that the RH is less than 40% (apart from RF05_1 and RF05_2). Samples with an RH of 40% could have considerable mass concentrations of water. This would introduce bias in optical closure calculations, but this bias is not assessed. This limitation of the study should be spelled out clearly to the reader.

P5 L15. I would add the text ", including descriptions and validation data for the conversion of UHSAS and APS data to particle number size distributions". On first reading, I was left thinking that a description was missing of how the APS data were processed to give a geometric or mobility diameter (rather than the raw APS-measured aerodynamic diameter) as well as that for how the optical size was corrected for refractive index. It was only on second reading and double-checking the SI that I realized these descriptions are given.

P5 L23-24. This sentence ('particles were separated into BC-containing particles and BC-free particles') leads to confusion because, at the end of the previous paragraph, you state that you do not investigate external mixing. Please could the authors bring some clarity to these seemingly contradictory statements.

P6 L5-7. $m_{BC\text{-}free}$ is calculated using a volume fraction weighting mixing rule. What was the reasoning for constraining the choice of mixing rule for $m_{BC\text{-}free}$ to the volume fraction mixing? Why not try MG and BG mixing rules also?

P9 L5-6. 'if other absorbing components are present, the imaginary part ($k_{eBC}$) of $m_{eBC}$ would be greater…'. Not necessarily - if those additional components are absorbing, they could have a lower absorption than BC, and therefore lower $k_{eBC}$.

P9 L26-27. 'magnetite, whose refractive index is generally invariant between wavelengths 470 and 660 nm.' Reference is needed.

P10 L16-17. 'The $MR_{100}$ is the MR for particles with 100 nm BC core only, i.e. $D_c$ equals to 100 nm.' It is not clear why you calculated $MR_{100}$. Can you provide the reader with your motivations for calculating this quantity? After twice reading this manuscript, I do not understand the utility of this parameter.

P10 L18-19. 'To justify its feasibility, we derived $m_{eBC}$ for other flights using Eq. 9 and found the differences are less than 5% for both the real and imaginary parts'. This is very surprising as I would expect absorption coefficients to provide a very poor constraint on the real refractive index. Please could the authors provide more evidence, perhaps in the Supplement, to convince the reader that this discrepancy is <5%.

P10 L20. 'no $m_{eBC}$ value has been achieved in the retrieval.' What does this statement mean? Do the authors mean to say that no minimum location in the merit function defined by Eqn. 9 could be located within the prescribed search bounds of $n_{eBC}$ and $k_{eBC}$?

P12 L6. 'otherwise, higher $k_{eBC}$ values indicate larger contributions of absorbers other than BC at 660 nm.' This is not correct. First, other absorbing species may have a lower absorption strength that BC. Second, the choice of optical closure model effects the accuracy of the derived $k$ and it is not clear that these grey sphere models represent the particles in question. Third, no assessment of measurement biases is provided here, which would impact on the derived $k_{eBC}$; at the forefront of my mind are the known (up to ~50%) biases in absorption from filter-based absorption techniques, the impact of humidity on the scattering measurements, and the accuracy of the particle size distributions using the authors' chosen techniques and subsequent impacts on retrievals of $m$.

P13 L3-4. 'no value was attained with the CS model.' Why was no value attained? It is not clear why this is the case.

P13 L11. 'however, these values are still smaller than those for RF05_3.' This statement is misleading and is not comparing like-for-like. The value from Taylor et al. (2020) is a campaign average value rather than a single leg of a single flight (as is quantified by the value for RF05_3). Moreover, if the reported uncertainties for RF05_3 and Taylor *et al*. values are considered, then the $E_{Abs}$ match to within statistical uncertainty.

P13 L14. '$MR_{100}$ was equal to 5.3'. I do not follow the utility of this parameter.

P13 L16. 'implying that the remaining 41% of absorption was contributed by other absorbers'. This statement is not suitably justified by the discussion, because no assessment of the impact of measurement biases is provided (which I suspect are very large indeed), while the optical properties may not be described well for these particles by a grey sphere or core-shell Mie model. More justification/discussion of these confounding factors is needed, along with moderation of language used to reflect that the role of other absorbers is a hypothesis.

P13 L16-17. 'This huge absorption from other absorbers…'. This is strong language; now the authors are asserting with certainty that the model-measurement discrepancy is real and is attributed to other absorbers. The authors must moderate the language here to ensure that it is clear that the role of other absorbers is only a hypothesis.

P14 L8. The error in the quoted $MAC_{OA,470}$ is larger than the mean value, allowing for a negative MAC that is nonsensical. In addition to rectifying this issue, the authors should state what this uncertainty represents. Is it the precision in the retrieved $MAC_{OA,470}$? Or perhaps the standard deviation across the different model treatments?

P14 L9. 'slightly lower…'. I do not agree that this not lower. They are identical within the stated uncertainties.

P14 L10-11. 'suggesting BrC bleaching during transport considering our result is at a much longer wavelength'. Following my last comment above, this statement cannot be inferred from the reported MAC values, with the values reported in this paper highly uncertain. Indeed, the quoted uncertainty values are large even though the contributions from some error sources are ignored, such as model assumptions (i.e., grey sphere, invariant $m_{eBC}$ with wavelength) and measurement biases. This statement must be removed.

P14 L13. Uncertainty larger than value, giving nonsensical negative values for $MAC_{OA}$.

P14 L23-24. 'however, to the best of our knowledge, such high MAC for secondary BrC have not been documented'. This statement must be removed. It is at odds with the last sentence of the last paragraph, where you show that the MAC$_{OA}$ from your measurements are, within statistical uncertainty, the same as those reported by Taylor et al. (2020).

P17 L3. 'within the range expected for BC.' Please state the expected range and provide references.

P17 L11-12. 'The extreme underestimation from the AAE attribution method is mainly due to the fact that the AAE and absorption coefficients used in this method are not derived from BC alone…' This statement is not justifiable. It is not clear that the CS or homogeneous "grey sphere" models used by the authors are superior in their approach compared to the AAE-attribution model. For example, the "grey sphere" Mie models assume sphericity and homogeneous mixing, ignoring lensing effects as well as internal multiple scattering interactions. If I look at the Taylor (2020) paper that the authors refer to heavily, Taylor et al. show that the grey sphere models (using volume mixing, Bruggeman, and MG models) all overestimate the absorption. Instead, Taylor et al. show that two different semi-empirical models for MAC (parameterisations fitted to comprehensive numerical simulations of optical properties for coated aggregated using either T-Matrix or discrete dipole approximation models) perform very well. It is important that the authors recognise the limitations of their framework and they need to moderate their language; it is not supported by the current paper that the AAE method is inferior in predicting the true BrC contribution to absorption.

P18 L15: Error in MAC is bigger than mean value, which is nonsensical.

P18 L26. Authors assert that brown carbon attribution from AAE approach is underestimated, i.e. implying that the AAE attribution approach is inferior. This is not justified by current manuscript, which ignores major limitations and uncertainty estimates in the optical closure approach.

**Technical corrections**

P4 L7. The authors use the acronym 'FRP'. This acronym is not used again, so please remove.

P6 Eqn 6. The "2" in the chemical symbol for potassium sulfate needs to be subscripted.

P6 L10. The minus sign in "1.8 g cm-3" needs to be superscripted.

P8 Figure 3. The tick marks on the vertical axes for the adjacent absorption cross section plots (i.e., the columns corresponding to m$_{BC}$=1.95+0.79$i$, 2.26+1.26$i$) do not match up. Also, what are the diamond symbols on these plots?

P8 L12. "note" should be "noted".

P9 L14. 'range' should be 'ranges'.

P11 Figure 4. The positioning of tick marks on the lower horizontal scale confusing. Perhaps simply mark every interval of 0.2, because I think it is clear from the legend what the upper limit of the grey shaded region is.

P11 L4-7. This long sentence is difficult to read and does not make grammatical sense in places. Please revise.

P14 L7. The statement "and other flights from homogeneous models" is nonsensical and needs revising.

P16 L20. The authors state "This AAE" but do not provide the value. Please state the value for the AAE of BrC referred to in this sentence.

P18 L29. Please remove "in the meanwhile".

---

## Author Comment (AC1)

**Response to comments on "Light Absorption by Brown Carbon over the South-East Atlantic Ocean" by Anonymous referee #1**

This study used comprehensive aircraft measurements to investigate the aged biomass burning plumes from south African transported to the Atlantic Ocean. The authors used optical closure between iterated refractive indices measured scattering-absorption at λ=660nm, to derive a proposed " effective refractive index of BC " in order to account for all BC absorption at multiple wavelengths. This study had a valuable dataset and could potentially contribute to the understanding on the aerosol absorption at this climatically important region. However I have a few concerns about the methods of this study, before it can be considered for publication.

We thank both reviewers for their interests in this work and for taking the time to provide detailed and constructive comments on our study. Some of the major comments raised by the reviewers have encouraged us to improve our method and add an uncertainty analysis section to the paper. Although the major results and conclusions of this study were not significantly changed, these revisions have improved the manuscript and made our method and results more robust.

In the text below, reviewer comments are in blue, our responses are in black, and manuscript revisions are in green. "P" refers to page number and "L" refers to the line number.

Major:

1) My main concern is about the nil absorption of brown carbon at 660nm. There are many studies stating OA could be absorbing at relatively long visible wavelength, particularly for less-volatile organics from biomass burning (Saleh et al. ES&T letters, 2018). These OA has a large molecular weight and more functionalized (hereby lower volatility) and may survive after transport, when more volatile species were evaporated with remaining more absorbing component to be transported to a longer distance, as evidenced by a recent field study (Liu et al., ACP 2021).

So it is very possible that the long visible absorption (e.g. 660nm) in your results contained some BrC. I would suggest it may not be necessary to really assume a nil absorption of BrC but using a combined positive k value for OA to feed the optical closure. See a study to derive the kOA by assuming some absorptivity for both externally and internally mixed OA in BC-containing particles (Liu et al., 2021 ES&T). This may lead to a more solid conclusion.

We thank the reviewer for pointing out the possibility of absorption at 660 nm by organic aerosols (OA) and suggesting to use a combined positive k value for OA. Biomass burning is one of the major sources of brown carbon (BrC) (Pósfai et al., 2004; Sedlacek III et al., 2018). For example, tar balls (Pósfai et al., 2004; Li et al., 2003), a common byproduct of wildfires, can absorb at long visible wavelengths (Hand et al., 2005; Chakrabarty et al., 2010; Hoffer et al., 2006). The

number fraction of tar balls can be ~36 % in hours-old plumes and decreases with aging (Pósfai et al., 2004; Hand et al., 2005; Sedelacek et al., 2018). We observed tar ball-like particles on filters from flight RF10 and RF11 with the number fraction smaller than 5%. Tar balls were not observed on filters of other flights. This low number fraction of tar ball-like particles is consistent with the highly aged (>6 days) aerosols in this study. However, the absorptivity of tar balls at 660 nm is unknown as the value of the imaginary refractive index of OA ($k_{OA}$) can vary over several orders of magnitude (Saleh et al., 2018). In addition, for aged aerosols, the photobleaching of BrC that occurs during transport will weaken its absorptivity. Table 1 summarized the $k_{OA}$ at the long visible wavelengths, together with their aging time from both experiments and field studies. To the best of our knowledge, there is no report in the literature about BrC absorptivity at long visible wavelengths for plumes with aging time longer than 6 days.

Table 1. Summary of imaginary refractive indices $k_{OA}$ of brown carbon at long visible wavelengths. The sources, aging time, and detected wavelengths are given as well.

| $k_{OA}$ | Sources | Aging time | Wavelength (nm) | Reference |
|---|---|---|---|---|
| 0.005 | Biomass smoke | Days-old smoke[a] | 650 | Kirchstetter et al. (2004) |
| 0.02 | Forest fire | Two or more days (tar balls dominated periods) | 632 | Hand et al. (2005) |
| 0.0014 | Tar balls, chamber study, Alaskan duff | Approx. 30 min | 780 | Chakrabarty et al. (2010) |
| 0.004 | Tar balls, chamber study, Ponderosa pine | Approx. 30 min | 780 | Chakrabarty et al. (2010) |
| 0.005-0.028 | Biomass burning emissions, chamber study | 1 hour after UV exposure | 660 | Saleh et al. (2013) |
| 0.15 | Tar balls, tar-water emulsion | Aged at 650 ℃ | 652 | Hoffer et al., (2016) |
| * | Tar balls, wildland fires | Hours-old plumes | - | Sedlacek III et al. (2018) |
| 0.03 | residential wood burning, flaming | Hours-old plumes | 781 | Liu et al. (2021) |
| ~0.001 | residential wood burning, smoldering | Hours-old plumes | 781 | Liu et al. (2021) |

[a] samples collected by the University of Washington's Convair-580 research aircraft in southern Africa in the SAFARI 2000 field study. Flight tracks are largely above southern Africa continent

(Sinha et al., 2003); therefore, the samples are much younger than those in ORACLES samples analyzed in our study, which were measured over the southeast Atlantic.

* no exact value was given; Mie calculations show consistency with weak light absorbance in Hand et al. (2005) and Chakrabarty et al. (2010) by tar balls.

To estimate the absorptivity of OA at 660 nm, we followed the method in Liu et al. (2021) and calculated the $k_{OA}$ at 660 nm for each flight. We assumed that any absorption not attributed to BC was solely from OA. A refractive index of BC of 1.95+0.79i was used; the real refractive index of OA was set to be 1.55. The particle number size distribution (PNSD) of BC-containing particles and BC-free particles are calculated with the BC core size distribution and coating thicknesses from SP2 and the PNSD of total aerosols from UHSAS and APS. The refractive index of BC-free particles are assumed to be the same as the non-BC components in the BC-containing particles. Detailed calculations of different models can be found in the supplement of the manuscript. Results of $k_{OA}$ at 660 nm are shown below.

[Figure]

Figure. 1 (a) The imaginary refractive index $k_{OA}$ at 660 nm and (b) the aerosol/plume age for each flight. The aerosol/plume age was modelled with a two-week forecast using the Weather Research and Aerosol Aware Microphysics (WRF-AAM) model.

As shown in Fig. 1, the $k_{OA}$ at 660 nm is lowest for RF11 and highest for RF05_1 and RF05_2 (Fig. 1a). Aerosols from flight RF11 are the least aged with an aerosol/plume age of 6.8±0.4 days, and the most aged are aerosols from flights RF05_1 and RF05_2, with the aerosol/plume age longer than 11 days (Fig. 1b). BrC experiences photolysis and photochemical bleaching during transport, processes which would lessen absorbance considerably (Laskin et al., 2015). Thus, the $k_{OA}$ values for aerosols in RF05_1 and RF05_2 with 4 more days aging time are expected to be much smaller than that of RF11. Moreover, the volatility of particles in our study is inconsistent with this increase of $k_{OA}$ with age as shown in Fig. 1, which we will further elaborate on below.

We thank the reviewer for bringing up Liu et al.'s (2021c) research on the evaporation of semivolatile organics during the uplift from the surface through the convective mixing and the possibility of lower-volatile aerosols to be transported to a longer distance. However, the TEM

analysis of the sampled aerosols suggests opposite results, in that we noted less organics on filters associated with more aged plumes, indicating more volatile aerosols, which is consistent with other companion studies such as Sedlacek et al. (In prep) and Dobracki et al. (In prep). Further, increasingly rounded and viscous organics on filters sampled from less aged plumes suggests an increase in aerosol volatility during transport (Dang et al., 2021). Possible explainations may relate to UV exposure and photooxidation that break down oligomers and lead to fragmentation of organic chains (refer to Section 3.2.1 in Dang et al. (2021) for more details). Higher absorbance of BrC is associated with an increase in molecular size and decrease in volatility (Saleh et al., 2018), thus, more aged aerosols appear to be more volatile in this study and are expected to be less absorbing, inconsistent with the trend of $k_{OA}$ in Fig. 1a.

Therefore, the variation of $k_{OA}$ with plume age as shown in Fig. 1 seems unreasonable considering the aging time and volatility. Particles from RF11, which are the least aged and less volatile, are expected to show the strongest OA absorptivity. This challenges the hypothesis that all absorption at 660 nm are solely contributed by BrC besides BC; other absorbers at long visible are expected to present. However, it is difficult to quantify the ratio of the absorptivity between BrC and other absorbing materials at 660 nm based on our current measurements. Therefore, we took the $k_{OA}$ at 660 nm of 0.003±0.001 from homogeneous models (as showed in the paper, CS model is not suitable for RF11) for RF11 as the upper bound of $k_{OA}$ for other flights, and calculated the $m_{eBC}$ and BrC absorption coefficients at 470, 530, and 660 nm. Results showed that even for the thickest case of RF10, the differences of BrC absorption coefficients at 3 wavelengths are all smaller than 9 %. Therefore, we kept our assumption that the BrC to total absorption at 660 nm can be neglected in the calculation, and added an evaluation the impact of this assumption as a new section into the revised version of the supplement.

**S2 Evaluation of the assumption of zero absorption of BrC at 660 nm**

[Figure]

Figure. S1 (a) The imaginary refractive index $k_{OA}$ at 660 nm and (b) the aerosol/plume age for each flight. The aerosol/plume age was modelled with a two-week forecast using the Weather Research and Aerosol Aware Microphysics (WRF-AAM) model.

To support our assumption of negligible BrC absorption at 660 nm, we assumed the absorption not attributed to BC at 660 nm to be solely from OA, and calculated the imaginary part of the refractive index of OA, $k_{OA}$, at 660 nm with different the VM, BG, MG, and CS models, following the method in Liu et al. (2021). A refractive index of BC of 1.95+0.79i was used, which is an upper bound of commonly used BC refractive index and thereby yields an upper bound of BC absorption and a lower bound of BrC absorption. The real refractive index of OA was set to be 1.55. Results of $k_{OA}$ at 660 nm are shown in Fig. S2. The $k_{OA}$ at 660 nm is lowest for particles in RF11, the least aged with the aerosol/plume age of 6.8±0.4 days, and highest for RF05_1 and RF05_2, the most aged with the aerosol/plume age greater than 11 days. The increase of $k_{OA}$ for more aged aerosols seems to be in inconsistent with the fact that BrC experiences photolysis and photochemical bleaching during transport, which will thereby lessen its absorbance considerably (Laskin et al., 2015). Aqueous-phase aging may form light-absorbing oligomeric products, while we observe no obvious differences in the liquid water content during the transport of investigated flights. Furthermore, higher absorptivity of BrC is associated with an increase in molecular size and decrease in volatility (Saleh et al., 2018), therefore, more aged particles that tends to exhibit higher volatility in this study (Dang et al., 2021) are expected to show lower absorptivity than less aged ones, which is again inconsistent with the increase of $k_{OA}$ for more aged aerosols as shown in Fig. 1. Both the aging time and volatility challenges the counter-assumption that absorption at 660 nm is contributed solely by BrC except for BC; hence, we conclude that other absorbers are expected to exist.

We took the $k_{OA}$ of the least aged particles in RF11 as the upper limit of $k_{OA}$ for all flights and used it to calculate the $m_{eBC}$ and absorption coefficients of BrC at 660 nm. Results showed that the difference between the BrC absorption coefficients with $k_{OA}$ of RF11 and those calculated with the assumption of non-absorbing OA at 660 nm is smaller than 9 % for the extreme case of RF10 with the thickest coating. Therefore, we neglected the possible contribution of BrC at 660 nm in the retrieval of $m_{eBC}$.

2) Figure 4 is crucial and needs large improvement. Are these from all the straight-level runs from all flights? Error bars for each dot are required. It said only 6 dots had potential additional absorbing component, if so, why there were some "magnetite" but some without, these need to be discussed. There are still no solid evidences that the magnetite did exist and contributed to the absorption at long visible. I may suggest having a plot like "additional absorption besides BC" vs magnetite fraction, the former is from Fig. 4 and the latter is from Fig. 5. Additionally, the origin of magnetite is speculated but not evidenced, where did they come from? This needs exterior support.

Yes, all measurements are from straight and level runs. The following text has been added to the main text in Section 2.1 in P4 L18 in the marked-up version of the revised manuscript.

We investigated aerosol optical properties from straight and level runs during seven research flights (RF) from the ORACLES 2018 campaign.

We have also revised Figure 4 as suggested by the reviewer. Since the discussion of altitude is not included in Section 3.1, we removed the information of altitude from the figure. Rather than giving a single dot for each $k_{eBC}$ in the previous version, we now show a boxplot, which provides the information of the 10th, 25th, 50th, 75th, and 90th percentiles. The revised figure is shown below (P14 in the marked-up version of the revised manuscript).

[Figure]

Figure 4. Boxplot of $k_{eBC}$ (left y-axis) derived from different models and the MR values (orange open dots, right y-axis) for each flight. Error bars of MR represent 17% uncertainty. The right y-axis uses the log scale. Light and dark grey shaded region shows $k_{eBC} < 1.26$, the largest $k_{BC}$ value of BC determined by Moteki et al. (2010), and 0.79, the largest value of commonly used $k_{BC}$, respectively.

As for the presence of magnetite, we agree with the reviewer that it is a hypothesis. We know form EDX analysis that elemental Fe is present, while can not determine the Fe-related compound with our current measurements. However, we can be sure of the presence of Fe, as stong peak of Fe appear in the EDX analysis. We have examined all possible compounds of detected elements, and found that magnetite is the only compound that is not rare in the atmosphere and absorb strongly at 660 nm. Magnetite usually come from industrial activities, such as steel manufacturing, oil combustion, and vehicle emissions (Puffer et al., 1980; Liati et al., 2015a; Ito et al., 2018; Kurisu et al., 2019). As well, magnetite can be transformed at high temperatures from Fe(III) found in minerals such as goethite and hematite (Till et al., 2015; Ito et al., 2018), which happens to be the two most abundant forms of FeOx in African dust (Formenti et al., 2014). Thus, we hypothesize the conversion during biomass burning and emissions from industrial activities might be the two possible sources of magnetite. The text concerning magnetite is shown below (P15 L2-20 in the marked-up version of the revised manuscript).

… Therefore, we suspect higher $k_{eBC}$ values may be related to iron oxides (FeOx) that absorbs at 660 nm. We have examined all possible compounds of detected elements from EDX and found that magnetite is the only compound that is not rare in the atmosphere and also absorbs

strongly at 660 nm. Magnetite shows a strong and uniform absorptivity over the visible spectrum with reported imaginary refractive index ranging from 0.58 to 1.0 (Amaury et al., unpublished data, http://www.astro.uni-jena.de/Laboratory/OCDB/mgfeoxides.html; Ackerman and Toon, 1981; Zhang et al., 2015), and therefore can contribute to the high values of $k_{eBC}$ at 660 nm. Studies show that magnetite can be transformed from Fe(III) at high temperatures, such as goethite and hematite (Till et al., 2015; Ito et al., 2018), which happen to be the two most abundant forms of FeOx in African dust (Formenti et al., 2014). Therefore, we hypothesize that a part of the magnetite might be converted from Fe(III) during biomass burning. Additionally, magnetite can be emitted from anthropogenic activities, such as steel manufacturing, oil combustion, and engines and brakes of motor vehicles (Machemer, 2004; Liati et al., 2015b; Moteki et al., 2017; Kurisu et al., 2019). The significance of anthropogenic magnetite in radiative forcing has been investigated and highlighted in recent studies (Moteki et al., 2017; Ito et al., 2018; Lamb et al., 2021). Although there is no investigation on magnetite in sub-Saharan Africa yet, the industrial and motor vehicle emissions in Africa are likely to contribute magnetite. Furthermore, the pyrometallurgical process is a widely used extraction method in copper mining, a major industry in the central African Copperbelt (https://www.pyrometallurgy.co.za/PyroSA/, Vítková et al., 2010; Sikamo et al., 2016; Shengo et al., 2019), of which iron is a common unwanted slag element (Meter et al., 1999). Therefore, we suspect it might also contribute magnetite considering the high temperature of the pyrometallurgical process.

3) The optical closure of the partial dataset is performed on absorption-only and others are absorption-scattering. My concern is this may induce some discrepancies, especially when presenting both datasets on the same table. Why not using consistent absorption-only approach, at least they are consistent.

We thank the reviewer for this suggestion. We followed the reviewer's advice and have redone the calculation with the absorption-only approach to maintain consistency. All related figures, numbers, and text have been revised accordingly.

Others:

1. There should be more details discussing about the determination of bulk mass ratio of coating over rBC core from the SP2 measurement.

   We thank the reviewer for the suggestion. We have revised Section 2.1 and moved the description of various instruments into the main text. We have added the determination of MR in the paragraph describing SP2, shown in P5 L27-31 in the marked-up version of the revised manuscript.

The mass ratio (MR) of non-BC substance to BC for BC-containing particles was determined by $MR = \frac{\sum_i (D_{p,i}^3 - D_{c,i}^3) * \rho_{BC-free}}{\sum_i D_{c,i}^3 * \rho_{BC}}$, where $D_p$ and $D_c$ represent the diameter of coated BC particle and BC core, respectively; the $i$ denotes the $i^{th}$ particle in the investigated time window. The determination of $\rho_{BC\text{-free}}$ will be discussed in Section 2.2.

**2. The title itself only includes brown carbon, though the authors also largely discussed about the possible absorption by iron.**

We do have lots of discussions of iron in the paper and hypothesize the presence of absorbing iron oxides in the investigated region. We have carefully examined whether to include iron in the title or not, and feel it might be better not include *potential iron oxides* in the title. But we do appreciate the reviewer for bringing this up.

3. It would be useful to briefly mention what effective refractive index of BC is in the abstract.

We thank the reviewer for the suggestion. We have revised as follows and in P1 L27-29 in the marked-up version of the revised manuscript.

… An effective refractive index of black carbon (BC), $m_{eBC}=1.95+ik_{eBC}$, that characterizes the absorptivity of all absorbing components at 660 nm wavelength was introduced to facilitate the attribution of absorption at shorter wavelengths, i.e. 470 nm.

4. Conclusions should include some discussions about evolution.

We thank the reviewer for the comment, we have added a comparison of BrC absorption to those in CLARIFY 2017, which implied a bleaching of BrC during transport. It is in P23 L24-27 in the marked-up version of the revised manuscript.

… Values from our results were generally comparable or slightly higher than the MAC$_{OA}$ of $0.31\pm0.09$ m$^2$ g$^{-1}$ and $R_{BrC}$ of ~11 % at 405 nm for highly aged aerosols from CLARIFY 2017, which were sampled downwind of ORACLES, while under a much longer wavelength, i.e. 470 nm, implying the bleaching of BrC during transport.

5. Page 2, line 3, what is the commonly used AAE? To what extent lower?

We thank the reviewer for bring this up. We have removed the sentence and added the following sentence to make it more clear as shown in P2 L5-7 in the marked-up version of the revised manuscript.

… Absorption attribution using the Bruggeman mixing Mie model suggests a minor BrC contribution of 4 % at 530 nm, while its removal would triple the BrC contribution to the total absorption at 470 nm obtained using the AAE attribution method.

6. The last sentence in the abstract needs rewriting, it is too long and a bit confusing.

We thank the reviewer for bring this up. We agree that the two "long wavelength" make the sentence confusing. We have revised it as follows and shown in in P2 L10-13 in the marked-up version of the revised manuscript.

Thus, it is recommended that the application of any optical properties-based attribution method use absorption coefficients at the longest possible wavelengths to minimize the influence of BrC, and to account for potential contributions from other absorbing materials.

7. Page 8, line 9, no need italic font.

We have removed italic font.

8. Page 10, line 5, MAC is mass absorption cross section, not coefficient.

We have changed it to mass absorption cross section.

9. Equation (9) why using sigma, but only one wavelength is used.

We thank the reviewer for pointing it out. We have removed sigma and added the wavelength 660 in the equation. One equation has been removed, so it is now Equation (8) and shown in P11 L7 in the marked-up version of the revised manuscript.

$$\chi^2(k_{eBC}) = (\frac{\sigma_{abs,660}^{cal} - \sigma_{abs,660}^{mea}}{\sigma_{abs,660}^{mea}})^2,$$

References

Liu, Q., et al.: Reduced volatility of aerosols from surface emissions to the top of the planetary boundary layer, Atmos. Chem. Phys., 21, 14749–14760, https://doi.org/10.5194/acp-21-14749-2021, 2021.

Liu D. et al.: Evolution of Aerosol Optical Properties from Wood Smoke in Real Atmosphere Influenced by Burning Phase and Solar Radiation, ES&T, 55(9), 5677–5688, 2021.

Saleh, R., Z. Cheng, and K. Atwi (2018), The Brown–Black Continuum of Light-Absorbing Combustion Aerosols, Environmental Science & Technology Letters, doi: 10.1021/acs.estlett.8b00305.

**Response to comments on "Light Absorption by Brown Carbon over the South-East Atlantic Ocean" by Anonymous referee #2**

**Summary and general comments**

The authors report spectroscopy measurements of aerosol optical properties from a research aircraft platform for biomass burning aerosols over the South East Atlantic Ocean. A modelling framework is developed for attributing the contribution of light absorbing organics (brown carbon, BrC) to the overall aerosol light absorption. This framework exploits either Core-Shell or 'grey sphere' Mie models, with the latter utilising one of three different effective medium approximations to treat the refractive index. The manuscript is interesting, uses established measurement methods, and adds to the debate surrounding the suitability of Mie models for treating aerosol optical properties. However, the authors assert conclusions that are not reinforced by their analysis, not least because there is scant treatment of systematic uncertainties in their measurements nor appreciation of the limitations imposed by their modelling assumptions. Considerable moderation of some of the claims of key outcomes is needed. Specific Comments are provided below in addition to Technical Corrections that need to be addressed.

We thank both reviewers for their interests in this work and for taking the time to provide detailed and constructive comments on our study. Some of the major comments raised by the reviewers have encouraged us to improve our method and add an uncertainty analysis section to the paper. Although the major results and conclusions of this study were not significantly changed, these revisions have improved the manuscript and made our method and results more robust.

In the text below, reviewer comments are in blue, our responses are in black, and manuscript revisions are in green. "P" refers to page number and "L" refers to the line number.

**Specific comments** (referring to the page "P" and line "L" numbers)

1. P1 L32. '$0.25\pm0.34$ m$^2$ g$^{-1}$': The error is larger than the mean value. The MAC cannot physically be negative, so this uncertainty is nonsensical. Either this uncertainty needs reconsidering, or some explanation of the significance of a negative MAC is needed.

   We thank the reviewer for pointing out this issue. It was caused by our former retrieval method that gave one $m_{eBC}$ value for each flight. We have revised our method by retrieving $m_{eBC}$ values every 10 seconds, consistent with the time resolution of measurements used in this study, and recalculated the MAC. The uncertainty has been largely reduced in this revised method; the average and standard deviation are now $0.30\pm0.27$ m$^2$ g$^{-1}$. We have updated all related values and figures accordingly.

2. P5 L11-12. '...except for the Neph, whose humidity is not controlled.' The lack of control of the humidity inside the nephelometer is a big issue in terms of the potential for large systematic uncertainties in measured optical properties that then impact on the later retrievals of $m_e$BC and attribution of absorption to brown carbon. There is no discussion in manuscript of humidity impacts on scattering coefficients and subsequent optical closure calculations. Indeed, no details on the RH in the Neph are provided for the different flights. The Supporting Information just states that the RH is less than 40% (apart from RF05_1 and RF05_2). Samples with an RH of 40% could have considerable mass concentrations of water. This would introduce bias in optical closure calculations, but this bias is not assessed. This limitation of the study should be spelled out clearly to the reader.

We thank the reviewer for pointing out the concern that 40 % RH may still involve some water that contribute to particle scattering. Strictly speaking, deliquescent substances can be regarded as dry at a relative humidity (RH) lower than the efflorescence relative humidity (ERH) (not the deliquescence relative humididy in case of hysteresis behavior); while for particles like ammonium nitrate whose ERH are extremely low (<5%), they grow continuously with RH and are very difficult to be completely dry (Hu et al., 2011). Thus, we agree with the reviewer that it is difficult to completely dry up particles in the measurement. We attempted to estimate the influence of RH on scattering coefficients by comparing the measured scattering coefficients with the dry scattering coefficients calculated with particle number size distribution and chemical composition under dry conditions. Figure S1 in the former version of supplement was used to illustrate the comparison of the measured scattering coefficients and calculated dry scattering coefficients. Besides RF05_1 and RF05_2, measured scattering coefficients from other flights all showed good agreement with those calculated under dry conditions, suggesting the contributions of water to the total scattering might be neglected in these flights. However, the inconsistency that partial dataset been constrained by absorption and others by both absorption and scattering may induce some discrepancies. Therefore, we revised our calculation method by assuming $m_{eBC}$=1.95+$k_{eBC}$ and using the absorption-only method to constrain $k_{eBC}$ as suggested by the first reviewer. This revised method assures the consistency among flights and also spelled out the influence of scattering coefficient. We also evaluated the assumption of constant real effective refractive index to absorption coefficients and the following paragraph has been added to the main text in P10 L21-25 in the marked-up version of the revised manuscript.

… The real part of $m_{eBC}$ is held constant at 1.95, the upper bound of the values commonly used for BC (Bond and Bergstrom, 2006; Saleh et al., 2013; Liu et al., 2015; Kahnert and Kanngießer, 2020). It has only a minor influence on absorption calculations (Liu et al., 2021): sensitivity test of the calculated absorption to the real part of $m_{eBC}$ when the latter is varied from 1.75 to 2.26 shows that, at the extreme case of RF10 with the thickest coating in this study, the absorption increased less than 5 % at 660 nm.

All related figures, instrument descriptions, and text have been changed accordingly in the revised manuscript.

3. P5 L15. I would add the text ", including descriptions and validation data for the conversion of UHSAS and APS data to particle number size distributions". On first reading, I was left thinking that a description was missing of how the APS data were processed to give a geometric or mobility diameter (rather than the raw APS-measured aerodynamic diameter) as well as that for how the optical size was corrected for refractive index. It was only on second reading and double-checking the SI that I realized these descriptions are given.

We thank the reviewer for this suggestion. We have revised the retrieval method, using the absorption-only method and hence do not need the data from UHSAS and APS now. We have removed these two instruments from Section 2.1 *Site and Instrumentation* and moved the instrument descriptions to the main text, which can be found in P5 L18-P6 L18 in the marked-up version of the revised manuscript.

4. P5 L23-24. This sentence ('particles were separated into BC-containing particles and BC-free particles') leads to confusion because, at the end of the previous paragraph, you state that you do not investigate external mixing. Please could the authors bring some clarity to these seemingly contradictory statements.

We thank the reviewer for this comment. We have revised the method, focusing on BC-containing particles, and excluded the BC-free particles. This part has been revised as follows and can be found in P6 L24-P7 L1 in the marked-up version of the revised manuscript.

We investigated the sensitivity of our closure simulations to four different models for the BC-containing particles – the ideal core-shell (CS) model and three homogeneous mixing models: 1) the volume mixing (VM) model, 2) the Maxwell-Garnett (MG) model, and 3) the Bruggeman (BG) model. The number size distribution of BC-containing particles was obtained from the BC core size distribution and the BC 2-D size and mixing state (i.e. coating thickness) distribution from SP2. Detailed descriptions and inputs of the four models can be found in Section S1 in the supplement.

5. P6 L5-7. $m_{BC\text{-free}}$ is calculated using a volume fraction weighting mixing rule. What was the reasoning for constraining the choice of mixing rule for $m_{BC\text{-free}}$ to the volume fraction mixing? Why not try MG and BG mixing rules also?

We thank the reviewer for this comment. The mixing state of BC is an important factor that influences the absorption of particles, thus various mixing assumptions have been applied to describe the mixing state of BC-containing particles in literature (Mackowski

and Mishchenko, 2011; Liu et al., 2017; Taylor et al., 2020). For example, the core-shell model assumes a BC core in the center and BC-free components as coating. However, the refractive index of BC-free components only has a minor effect on the absorption calculation (Liu et al., 2021a), and researchers usually do not apply these mixing rules to BC-free components. Most of the time, a fixed value (usually around 1.5) is set as the refractive index of the BC-free part (Liu et al., 2021a; Saleh et al., 2013). The $m_{\text{BC-free}}$ in our study is calculated with the AMS measurements and is 1.52 on average, consistent with the values used in previous studies (Liu et al., 2021a; Saleh et al., 2013). We have added the following description in Section 2.2 in P7 L15-18 in the marked-up version of the revised manuscript.

The real part of the refractive index of OA varies from 1.35 to 1.7 (Lack and Cappa, 2010; Liu et al., 2013; Saleh et al., 2014; Moise et al., 2015); 1.55 is used in this study. OA is assumed to be non-absorbing in the calculation. The average and standard deviation of calculated $m_{\text{BC-free}}$ is 1.52±0.015, consistent with those used in BrC studies in literature (Saleh et al., 2013; Liu et al., 2015, 2021a).

6. P9 L5-6. 'if other absorbing components are present, the imaginary part ($k_{\text{eBC}}$) of $m_{\text{eBC}}$ would be greater...'. Not necessarily - if those additional components are absorbing, they could have a lower absorption than BC, and therefore lower $k_{\text{eBC}}$.

We thank the reviewer for the comment. In the retrieval process, the $m_{\text{eBC}}$ is calculated with the BC core size distribution, coating thickness distribution, and the absorption coefficient at 660 nm. It accounts for the absorptivity of all components that absorb at 660 nm. The $m_{\text{eBC}}$ value is varied until the differences between the calculated and measured absorption coefficients are minimised. Therefore, if BC is the only absorber at 660 nm, $m_{\text{eBC}}$ will be the same as $m_{\text{BC}}$; however, if there is another absorber at 660 nm, no matter how small its absorptivity is, this absorptivity will be "added" to that of BC, making $m_{\text{eBC}}$ larger than $m_{\text{BC}}$. We understand the reviewer's concern that the $m_{\text{eBC}}$ might be calculated as e.g. a volume average of all absorbers; while in our retrieval process, their absorptivity are all attributed to BC. That is also the reason why we name $m_{\text{eBC}}$ the effective refractive index of BC.

7. P9 L26-27. 'magnetite, whose refractive index is generally invariant between wavelengths 470 and 660 nm.' Reference is needed.

We thank the reviewer for pointing it out, the reference has now been added in P12 L3-5 in the marked-up version of the revised manuscript.

…magnetite, whose refractive index is also mostly invariant between wavelengths 470 and 660 nm (Amaury et al., unpublished data, http://www.astro.unijena.de/Laboratory/OCDB/mgfeoxides.html; Ackerman and Toon, 1981; Zhang et al., 2015).

8. P10 L16-17. 'The $MR_{100}$ is the MR for particles with 100 nm BC core only, i.e. $D_C$ equals to 100 nm.' It is not clear why you calculated $MR_{100}$. Can you provide the reader with your motivations for calculating this quantity? After twice reading this manuscript, I do not understand the utility of this parameter.

We thank the reviewer for bringing this up; we have now removed the parameter $MR_{100}$. The coating thickness varies with the size of BC core, and 100 nm BC core is usually very thick coated. Thus, $MR_{100}$, the MR for particles with 100 nm BC core, can generally represent an upper bound of the MR for a BC core size distribution. However, we noticed that the $MR_{100}$ is only mentioned twice in the paper, and both are with bulk MR. As bulk MR is enough for our discussion, the parameter $MR_{100}$ has now been removed.

9. P10 L18-19. 'To justify its feasibility, we derived $m_{eBC}$ for other flights using Eq. 9 and found the differences are less than 5% for both the real and imaginary parts'. This is very surprising as I would expect absorption coefficients to provide a very poor constraint on the real refractive index. Please could the authors provide more evidence, perhaps in the Supplement, to convince the reader that this discrepancy is <5%.

We thank the reviewer for pointing out the concern of the inconsistency between the two constrain methods. As replied above to comment 2, we have now revised our method, using absorption-only method to constrain $k_{eBC}$. The real part of the effective refractive index of BC ($n_{eBC}$), is set as 1.95. We have also investigated the sensitivity of absorption to $n_{eBC}$. Revised text can be found in P10 L21-25 in the marked-up version of the revised manuscript. Related texts and figures have been revised accordingly in Section 2.2 and 2.3.

10. P10 L20. 'no $m_{eBC}$ value has been achieved in the retrieval.' What does this statement mean? Do the authors mean to say that no minimum location in the merit function defined by Eqn. 9 could be located within the prescribed search bounds of $n_{eBC}$ and $k_{eBC}$?

The predicted absorption coefficients with the core-shell model are much lower than measurements. There is a criteria, 'Values of $\chi^2$ greater than 0.1 are excluded', in Section 2.3 right after equation 8. No absorption cofficients calculated with the core-shell model meet this criteria within our $k_{eBC}$ search bound, i.e. 0.3-2.6. We have calculated the absorption efficiency $Q_{abs}$ varying the real part $n$ from 1 to 3 and imaginary $k$ from 0 to 10. Figure 1a illustrates the variation of the absorption efficiency $Q_{abs}$ with $n$ and $k$ at 660 nm. The particle was assumed to have a 100 nm BC core and 43 nm coating thickness following RF05_3. The largest $Q_{abs}$ is 1.25 at $k$ of 2.3 as shown in Fig. 1a. The variation of the largest $Q_{abs}$ at 660 nm with particle size is shown in Fig. 1b. The coating thickness for each BC

core diameter was taken as the average of those in RF05_3. The largest $Q_{abs}$ is 1.48 from Fig. 1b. We used the largest $Q_{abs}$ of 1.48 to calculate the absorption coefficients for RF05_3, while they are still much lower than the measurements and hence do not meet the $\chi^2$ criteria. We have revised the sentence to make it clearer, which can be found in P12 L25-29 in the marked-up version of the revised manuscript.

No $m_{eBC}$ for RF05_3 was obtained with the CS model because the criteria of $\chi^2$ smaller than 0.1 was not met, as the absorption coefficient calculated with the CS model using the largest possible $m_{BC}$ (i.e. 2.26+1.26i) was only half of the measurement for RF05_3 (Fig. S2).

[Figure]

Figure 1. (a) The variation of $Q_{abs}$ with the real ($n$) and imaginary ($k$) part of the refractive index of BC at 660 nm for particle with 100 nm BC core and 43 nm coating thickness following RF05_3. (b) The variation of the largest possible absorption efficiency $Q_{abs}$ with the BC core diameter at 660 nm. The coating thickness is taken from RF05_3.

11. P12 L6. 'otherwise, higher $k_{eBC}$ values indicate larger contributions of absorbers other than BC at 660 nm.' This is not correct. First, other absorbing species may have a lower absorption strength that BC. Second, the choice of optical closure model effects the accuracy of the derived $k$ and it is not clear that these grey sphere models represent the particles in question. Third, no assessment of measurement biases is provided here, which would impact on the derived $k_{eBC}$; at the forefront of my mind are the known (up to ~50%) biases in absorption from filter-based absorption techniques, the impact of humidity on the scattering measurements, and the accuracy of the particle size distributions using the authors' chosen techniques and subsequent impacts on retrievals of $m$.

We thank the reviewer for this comment. The detailed response to the reviewer's first concern of absorbers with lower absorptivity than BC can be found in the response to

comment 6. In short, the $m_{eBC}$ is retrieved from the absorption from PSAP and the BC core and thickness distribution from SP2, and is therefore the "sum" of the absorptivity at 660 nm of all absorbing components. Regarding the second concern of the representativeness of these models, we agree with the reviewer that every model is an approximation and involves lots of assumptions. This is also the reason that motivated us to test these four models and evaluate their results in this study. The reason why we have not applied T-matrix or discrete dipole approximation model is that they requires lots of assumptions of particle geometry without much information to guide us. We have added this discussion to P6 L20-24 in the marked-up version of the revised manuscript.

Most of the particles were found to be nearly spherical from the images obtained by TEM, with >70 % of the particles having aspect ratios (the largest and smallest characteristic sizes of arbitrarily shaped particle) smaller than 1.5. Therefore, we applied Mie theory to determine aerosol optical properties. Models such as T-matrix or discrete dipole approximation model were not used as they would require a large number of free parameters (Scarnato et al., 2013; He et al., 2016).

As to the third point, we thank the reviewer for pointing out the lack of uncertainty analysis in this study. We have now added the uncertainty analysis to make this study and results more robust. As responded earlier to comments 1 and 2, we have revised the method and spelled out the influence of scattering measurements under inconsistent RHs, therefore, the UHSAS measurements are no longer needed. The measurements we use now are mainly from PSAP, SP2, and AMS. Descriptions of these instruments have now been moved from the supplement to the main text.

We have also added a section discussing the uncertainty of the results in our study using the Monte-Carlo uncertainty analysis in the supplement. We can not determine these uncertainties analytically due to the calculation complexity. Following the approach in Taylor et al. (2020), we generated arrays of scale factors and performed the calculation for 10000 times randomly using the Monte Carlo approach. Results show that the uncertainty of $k_{eBC}$ was ~24 % using the homogeneous models and 35 % with the CS model. The following section has been added to the supplement.

**S3 Uncertainty analysis**

The uncertainty of the imaginary part of the effective refractive index of BC ($k_{eBC}$) and absorption coefficients calculated with $m_{eBC}$ and $m_{BC}$ of 1.95+0.79i at 470, 530, and 660 nm ($\sigma_{abs,eBC}$ and $\sigma_{abs,BC}$) were estimated using the Monte Carlo uncertainty analysis. It is applied because of the complexity of the retrieval process of $k_{eBC}$ and the calculation of $\sigma_{abs,eBC}$ and $\sigma_{abs,BC}$. The uncertainty of absorption coefficients of BrC at 470 and 530 nm ($\sigma_{abs,470,BrC}$ and $\sigma_{abs,530,BrC}$), the mass absorption cross section of OA at 470 nm ($MAC_{OA,470}$),

and the contribution of BrC to the total absorption at 470 nm ($R_{BrC,470}$) were then calculated analytically.

Following the approach in Taylor et al. (2020), we used the uncertainty in each input variable to generate an array of scale factors to represent the variability of the variable may have when measured a large number of times. Specifically, we first generated an array of scale factors that follows the Gaussian distribution with a mean of 1 and a standard deviation of the uncertainty for each input variable. The array of scale factors was then multiplied by the corresponding input variable to generate an array of variables, representing the possibility of this input variable if it were measured a large number of times, 10000 is used in this study. Variables considered in this analysis include the BC core size distribution, BC coating thickness, absorption coefficients, and OA mass concentration (Fischer et al., 2010). For the BC coating thickness and OA mass concentration, we used the conservative value of 2-σ uncertainty (Bahreini et al., 2009; Taylor et al., 2020). Input variables and corresponding uncertainties are shown in Table S1. An uncertainty of 4% was given to the real part of refractive index of the coating following Taylor et al. (2020).

Table S1. Uncertainty of input variables of the Monte-Carlo uncertainty analysis, corresponding instruments were also given.

|  | Uncertainty | Instrument |
|---|---|---|
| BC core mass | 20% | SP2 |
| Coating thicknesses | 22% | SP2 |
| Measured absorption coefficients | 20% | PSAP |
| OA mass | 38% | AMS |

The results of uncertainty analysis are shown in Table S2. The uncertainties from VM, MG, and BG models are very close. The uncertainty of $k_{eBC}$ obtained from homogeneous models and the CS model was 24 % and 35 %, respectively. The uncertainty of $R_{BrC,470}$ and $MAC_{OA,470}$ was ~35% and 48%, respectively. The high uncertainty of $MAC_{OA,470}$ mainly results from the large uncertainty of OA mass measured by the AMS. While all $MAC_{OA}$ would be subject to this large uncertainty if the OA mass was determined by AMS.

Table S2. Monte Carlo relative standard deviations of $k_{eBC}$, $\sigma_{abs,eBC}$, and $\sigma_{abs,BC}$, and uncertainties of $\sigma_{abs,BrC}$, $R_{BrC,470}$, and $MAC_{OA,470}$ with different optical models.

|  | VM | MG | BG | CS |
|---|---|---|---|---|
| $k_{eBC}$ | 0.24 | 0.24 | 0.24 | 0.35 |
| $\sigma_{abs,660,eBC}$ | 0.22 | 0.23 | 0.23 | 0.24 |
| $\sigma_{abs,530,eBC}$ | 0.21 | 0.22 | 0.22 | 0.23 |

| | | | |
|---|---|---|---|
| $\sigma_{abs,470,eBC}$ | 0.21 | 0.21 | 0.21 | 0.23 |
| $\sigma_{abs,660,BC}$ | 0.56 | 0.56 | 0.56 | 0.51 |
| $\sigma_{abs,530,BC}$ | 0.53 | 0.53 | 0.55 | 0.47 |
| $\sigma_{abs,470,BC}$ | 0.51 | 0.51 | 0.51 | 0.45 |
| $\sigma_{abs,530,BrC}$ | 0.29 | 0.30 | 0.30 | 0.30 |
| $\sigma_{abs,470,BrC}$ | 0.29 | 0.29 | 0.29 | 0.30 |
| $R_{BrC,470}$ | 0.35 | 0.35 | 0.35 | 0.36 |
| $MAC_{OA,470}$ | 0.48 | 0.48 | 0.48 | 0.48 |

Following the review's advice, we have revised the sentence in P14 L12-13 in the marked-up version of the revised manuscript.

…higher $k_{eBC}$ value indicates larger absorptivity of absorbers at 660 nm.

12. P13 L3-4. 'no value was attained with the CS model.' Why was no value attained? It is not clear why this is the case.

We thank the reviewer for this comment. Please refer to the response to comment 10.

13. P13 L11. 'however, these values are still smaller than those for RF05_3.' This statement is misleading and is not comparing like-for-like. The value from Taylor et al. (2020) is a campaign average value rather than a single leg of a single flight (as is quantified by the value for RF05_3). Moreover, if the reported uncertainties for RF05_3 and Taylor *et al*. values are considered, then the $E_{Abs}$ match to within statistical uncertainty.

We thank the reviewer for pointing this out. We have revised "smaller than" to "generally comparable or smaller than". The $E_{abs}$ values are generally comparable and the $MAC_{BC}$ in CLARIFY is smaller than those in RF05_3. It has been revised in P16 L10-13 in the marked-up version of the revised manuscript.

Taylor et al. (2020) presented relatively high $MAC_{BC}$ values of 20±4, 15±3, and 12±2 $m^2$ $g^{-1}$ at 405, 514, and 655 nm, respectively and an $E_{Abs}$ of 1.85±0.45 for CLARIFY 2017 (Cloud-Aerosol-Radiation Interaction and Forcing 2017 measurement campaign; Haywood et al., 2020), which are generally comparable or smaller than those for RF05_3.

14. P13 L14. '$MR_{100}$ was equal to 5.3'. I do not follow the utility of this parameter.

We have now removed the parameter $MR_{100}$. Please refer to the response to comment 8.

15. P13 L16. 'implying that the remaining 41% of absorption was contributed by other absorbers'. This statement is not suitably justified by the discussion, because no assessment

of the impact of measurement biases is provided (which I suspect are very large indeed), while the optical properties may not be described well for these particles by a grey sphere or core-shell Mie model. More justification/discussion of these confounding factors is needed, along with moderation of language used to reflect that the role of other absorbers is a hypothesis.

We thank the reviewer's comment. We have added evaluation of the model and the BC refractive index and revised our language. The sentence has been revised, as shown in P16 L17-L25 in the marked-up version of the revised manuscript.

We calculated absorption coefficients for RF05_3 at 660 nm using the VM model, which usually yields a higher estimation of the measurements (Taylor et al., 2020), and a $m_{BC}$ of 1.95+0.79i, an upper bound of the BC refractive index (Liu et al., 2021). Uncertainties related to calculated absorption coefficients can be found in Section S3 in the supplement. The calculated absorption, which is very likely an overestimation of measurements, is only able to explain ~57 % of the measured absorption coefficient (Fig. S2), implying that the remaining amount of absorption which cannot be attributed to BC might result from other absorbers. This unexplained portion of absorption in this flight is much higher than those in other flights (e.g. Fig. 3 and Fig. S2).

16. P13 L16-17. 'This huge absorption from other absorbers...'. This is strong language; now the authors are asserting with certainty that the model-measurement discrepancy is real and is attributed to other absorbers. The authors must moderate the language here to ensure that it is clear that the role of other absorbers is only a hypothesis.

We thank the reviewer's comment and acknowledge that these strong language should be avioded. We have revised related sentences in the discussion, which can be found in P16 L23-L32 in the marked-up version of the revised manuscript.

…This unexplained portion of absorption in this flight is much higher than those in other flights (e.g. Fig. 3 and Fig. S2). Back-trajectories show that air parcels in RF05_3 passed by the Copperbelt in Zambia, where the pyrometallurgical process in copper mining might contribute large amounts of magnetite. … Therefore, we suspect this unexplained absorption in RF05_3 may also be related to absorbing titanium compounds.

17. P14 L8. The error in the quoted $MAC_{OA,470}$ is larger than the mean value, allowing for a negative MAC that is nonsensical. In addition to rectifying this issue, the authors should state what this uncertainty represents. Is it the precision in the retrieved $MAC_{OA,470}$? Or perhaps the standard deviation across the different model treatments?

The larger error is mainly caused by using a constant $m_{eBC}$ for the whole flight. We have revised the retrieval method and redone the calculation. The average and standard deviation is now 0.30±0.27. Please also refer to the response to comment 1. The sentence has been revised as follows, which are shown P17 L15-17 in the marked-up version of the revised manuscript.

…with the values (average±standard deviation) of $MAC_{OA,470}$ for RF05_3, RF06_1, RF06_2, RF10, and RF11 ranging from 0.30±0.27 m$^2$ g$^{-1}$ to 0.68±0.08 m$^2$ g$^{-1}$.

18. P14 L9. 'slightly lower...'. I do not agree that this not lower. They are identical within the stated uncertainties. P14 L10-11. 'suggesting BrC bleaching during transport considering our result is at a much longer wavelength'. Following my last comment above, this statement cannot be inferred from the reported MAC values, with the values reported in this paper highly uncertain. Indeed, the quoted uncertainty values are large even though the contributions from some error sources are ignored, such as model assumptions (i.e., grey sphere, invariant $m_{eBC}$ with wavelength) and measurement biases. This statement must be removed.

We agree with the reviewer that the result from Taylor et al. (2020) are within the range of our results. We have removed 'slightly lower' and 'suggesting BrC bleaching during transport considering our result is at a much longer wavelength' as suggested by the reviewer. The sentence has been revised in P17 L17-P18 L2 in the marked-up version of the revised manuscript.

… This result at 470 nm is generally comparable to the value of 0.31±0.09 m$^2$ g$^{-1}$ at 405 nm for highly aged aerosols sampled downwind of ORACLES in CLARIFY 2017 campaign (Taylor et al., 2020).

19. P14 L13. Uncertainty larger than value, giving nonsensical negative values for $MAC_{OA}$.

The larger error is mainly caused by using a constant $m_{eBC}$ for the whole flight. We have revised the retrieval method and redone the calculation. Please also refer to responses to comment 1 and 17.

20. P14 L23-24. 'however, to the best of our knowledge, such high MAC for secondary BrC have not been documented'. This statement must be removed. It is at odds with the last sentence of the last paragraph, where you show that the $MAC_{OA}$ from your measurements are, within statistical uncertainty, the same as those reported by Taylor et al. (2020).

Based on the value of $MAC_{OA,470}$, we seperated the discussion in Section 3.2 into two parts, one for flights RF05_3, RF06_1, RF06_2, RF10, and RF11 and the other for flights

RF05_1 and RF05_2. The reason of discussing flights RF05_1 and RF05_2 seperately is because their unexpected high $MAC_{OA,470}$ values. The sentence that the reviewer mentioned were for $MAC_{OA,470}$ of RF05_1 and RF05_2, different from those for flights RF05_3, RF06_1, RF06_2, RF10, and RF11 discussed in the first paragraph. To clarify, we emphasized that the high MAC are those in RF05_1 and RF05_2 and the sentence has now been revised in P18 L15-18 in the marked-up version of the revised manuscript..

… Saleh et al. (2013) reported that secondary BrC can be more absorbing than primary BrC at short visible wavelengths; however, to the best of our knowledge, such high MAC like those in RF05_1 and RF05_2 for secondary BrC have not been documented (Kasthuriarachchi et al., 2020).

21. P17 L3. 'within the range expected for BC.' Please state the expected range and provide references.

We thank the reviewer for the comment. The sentence has now been revised in P21 L30 in the marked-up version of the revised manuscript..

…which is just below unity and within the range expected for BC (Liu et al., 2018; Taylor et al., 2020).

22. P17 L11-12. 'The extreme underestimation from the AAE attribution method is mainly due to the fact that the AAE and absorption coefficients used in this method are not derived from BC alone...' This statement is not justifiable. It is not clear that the CS or homogeneous "grey sphere" models used by the authors are superior in their approach compared to the AAE-attribution model. For example, the "grey sphere" Mie models assume sphericity and homogeneous mixing, ignoring lensing effects as well as internal multiple scattering interactions. If I look at the Taylor (2020) paper that the authors refer to heavily, Taylor et al. show that the grey sphere models (using volume mixing, Bruggeman, and MG models) all overestimate the absorption. Instead, Taylor et al. show that two different semi-empirical models for MAC (parameterisations fitted to comprehensive numerical simulations of optical properties for coated aggregated using either T-Matrix or discrete dipole approximation models) perform very well. It is important that the authors recognise the limitations of their framework and they need to moderate their language; it is not supported by the current paper that the AAE method is inferior in predicting the true BrC contribution to absorption.

We thank the reviewer for the comment. This paragraph is a discussion and comparison of the results of BrC contribution from two methods, the AAE attribution method and optical closure method. Both are with advantages and disadvantages. The AAE attribution method is widely used for its simplicity, while it requires the absorption cofficients and AAE used

in the calculation to be solely for BC (excluding the impact of other absorbers), which is usually very difficult to meet. The optical closure method can give an attribution of the absorption of various absorbers, while as the reviewer mentioned, it is subject to various uncertainties. For example, the homogeneous models are usually reported to overestimate BC absorption thus underestimate the BrC contribution, while the BrC contribution from the AAE attribution method is even smaller than those from the homogeneous models. Therefore, we discussed the possible reason of this small contribution. Using the BG model, we found absorption contribute 4% of absorption at 530 nm, which is subject to 15% uncertinty as shown in Section S3 in the supplement. We tried to exclude the 4% BrC in the absorption at 530 nm and recalculated the BrC contribution at 470 nm with the AAE attribution method and found the result becomes two times larger. Therefore, we conclude that if BrC is involved in the calculation of the AAE attribution method, even it may exist as a very small portion, it can still have a substantial impact on the BrC absorption estimation. Therefore, we recommend to use the absorption at the longest possible wavelengths to minimise the influence of BrC when using the AAE attribution method. In the revised version, we avoided words like underestimate or overestimate but directly compare the results from the two methods. We have revised this paragraph and moderated our language as follows. Changes can be found in P22 L7-P23 L2 in the marked-up version of the revised manuscript.

The AAE attribution method is widely used for its simplicity, while there is one caveat that need to be noted when applying this method. The absorption coefficients and AAE, in this case, $\sigma_{abs,530}^{mea}$, $\sigma_{abs,660}^{mea}$, and AAE$_{530/660}$, should be for BC only, without contributions from other absorbers, such as BrC (Lack and Cappa, 2010). However, it is usually very difficult to completely exclude the impact of these absorbers in the calculation. The homogeneous models are usually reported to overestimate BC absorption (Zhang et al., 2015a; Fierce et al., 2017; Taylor et al., 2020) and hence underestimate BrC absorption; while the values of the contribution of BrC determined by the AAE attribution method are even lower than those from the homogeneous methods, which we suspect might be partly attributed to the impact of absorbers besides BC. Therefore, we evaluated the absorption contribution at detected wavelengths by taking RF06_1 as an example. The contribution of BrC derived from the AAE attribution method is 2% for RF06_1, approximately a factor of five smaller than that from the optical closure method with the BG model, 11% (Fig. 7 and 8). As illustrated in Fig. 8, $\sigma_{abs,530}^{mea}$ includes the contribution of BrC, which accounts for 4 % of the total absorption, and that of absorbers beyond BC and BrC, which accounts for 21 %. Note this attribution is based on the BC absorption coefficient calculated from the BG model with $m_{BC}$=1.95+0.79i, which yields an upper bound of BC absorption (Liu et al., 2021). All calculated parameters are subject to uncertainties as discussed in Section S3 in the supplement. If we remove the 4% BrC from $\sigma_{abs,530}^{mea}$ in Eq. 14, the BrC contribution to absorption at 470 nm would increase to 6%, two times larger than 2%, indicating a

substantial impact of BrC on the result of the AAE attribution method even though the BrC absorption may only exist as a small portion. Thus, we recommend that application of any optical properties-based attribution method to use absorption coefficients at the longest possible wavelength to minimize the influence of BrC, and in the meanwhile, to account for potential contributions from other absorbing materials.

23. P18 L15: Error in MAC is bigger than mean value, which is nonsensical.

We thank the reviewer for pointing out this issue. It is caused by our previous retrieval method that gave one $m_{eBC}$ value for each flight. We have revised our method and redone the calculation. Please refer to responses to comment 1, 17, and 19. We have updated all the values, figures, and text accordingly.

24. P18 L26. Authors assert that brown carbon attribution from AAE approach is underestimated, i.e. implying that the AAE attribution approach is inferior. This is not justified by current manuscript, which ignores major limitations and uncertainty estimates in the optical closure approach.

We thank reviewer for this comment. The uncertainty analysis has now been added as a new section in the supplement. In the revised version, we avoided using the strong words like underestimate or overestimate. We have revised the part discussing these two methods (refer to the response to comment 22). Absorption attribution using the optical closure method with the BG model noted a small contribution (4%) of absorption by BrC at 530 nm, which is subject to 29 % uncertainty, and excluding this small BrC absorption would increase the BrC contribution at 470 nm by two times. Therefore, we regard that the BrC can have a substantial impact on the BrC absorption estimation in the AAE attribution method even though it may exist as a very small portion. We have revised this sentence and changes can be found in P24 L4-7 in the marked-up version of the revised manuscript.

… Absorption attribution using the Bruggeman mixing Mie model suggest a minor BrC contribution of 4 % at 530 nm for RF06_1, and its removal would triple the BrC contribution to the total absorption at 470 nm from 2 % to 6 % obtained using the AAE attribution method, suggesting a substantial impact of BrC to the result of AAE attribution method even though the BrC may only exist as a small portion.

**Technical corrections**

1. P4 L7. The authors use the acronym 'FRP'. This acronym is not used again, so please remove. P6 Eqn 6. The "2" in the chemical symbol for potassium sulfate needs to be

subscripted.

P6 L10. The minus sign in "1.8 g cm-3" needs to be superscripted.

We thank the reviewer for pointing these out. We have revised accordingly.

2. P8 Figure 3. The tick marks on the vertical axes for the adjacent absorption cross section plots (i.e., the columns corresponding to mBC=1.95+0.79*i*, 2.26+1.26*i*) do not match up. Also, what are the diamond symbols on these plots?

We thank the reviewer for pointing this out. We have revised the figure and added description of the boxplot in the figure caption. Since the revised retrieval method does not include scattering coefficients, we removed the plots for scattering coefficient. The figures has been revised as follows.

[Figure]

Figure 3: Modelled (blue and orange markers) and measured (green markers) absorption coefficients ($\sigma_{abs}$) at PSAP wavelengths for RF06_1. Variables are modelled with two commonly used $m_{BC}$ values (shown in the legend) using the CS, MG, BG, and VM models (specified on the top of each plot). OA is assumed to be non-absorbing with the refractive index $m_{OA}$ equal to 1.55+0i. The horizontal lines represent the median value, the boxes represent 25th to 75th percentile, the whiskers represent 1.5 inter-quartile range, and the diamonds represent outliers.

3. P8 L12. "note" should be "noted".

We thank the reviewer for the comment. We have changed to noted.

4. P9 L14. 'range' should be 'ranges'.

We thank the reviewer for the comment. We have changed to ranges.

5. P11 Figure 4. The positioning of tick marks on the lower horizontal scale confusing. Perhaps simply mark every interval of 0.2, because I think it is clear from the legend what the upper limit of the grey shaded region is.

We thank the reviewer for the comment. We have made revisions following the reviewer's suggestion. It has now been revised as follows.

[Figure]

Figure 4. Boxplot of $k_{eBC}$ (left axis) derived from different models and the MR values (orange open dots, right axis) for each flight. Error bars of MR represent 20 % uncertainty. The right y-axis uses the log scale. Light and dark grey shaded region shows $k_{eBC}$ smaller than 1.26, the largest $k_{BC}$ value determined by Moteki et al. (2010), and 0.79, the largest value of commonly used $k_{BC}$, respectively.

6. P11 L4-7. This long sentence is difficult to read and does not make grammatical sense in places. Please revise.

We thank the reviewer for the comment. The sentence has now been revised as follows. Changes can be found in P13 L16-18 in the marked-up version of the revised manuscript.

The use of CS model for RF10 was supported by its high MR value of 7.4, which has exceeded the lower bound of MR for the use of CS model proposed by Liu et al. (2017), although a fair number of BC particles homogeneously mixed with salts and OA were detected as well (Fig. 5).

7. P14 L7. The statement "and other flights from homogeneous models" is nonsensical and needs revising.

We thank the reviewer for the comment. The sentence has now been revised as follows. Changes can be found in P17 L14-16 in the marked-up version of the revised manuscript.

From Section 3.1, results of RF10 from the CS model and the results of other flights calculated with homogeneous models are more plausible…

8. P16 L20. The authors state "This AAE" but do not provide the value. Please state the value for the AAE of BrC referred to in this sentence.

We thank the reviewer for the comment. We have added the value and revised the sentence as follows. Changes can be found in P21 L4-15 in the marked-up version of the revised manuscript.

Therefore, we expect the AAE of BrC at 365/470 wavelength pair, $AAE_{365/470,BrC}$, derived from $\sigma_{abs,365,BrC}^{mea}$ and $\sigma_{abs,470,BrC}^{cal}$ (absorption coefficients of BrC at 365 and 470 nm) to be within the range of ~2-11 in literature (Laskin et al., 2015). Since $\sigma_{abs,365,BrC}^{mea}$ is unknown, we calculated $\sigma_{abs,365,extrapolate}$ by extrapolating the measured $\sigma_{abs,470}^{mea}$ to 365 nm with the measured $AAE_{470/530}$ and approximated the $AAE_{365/470,BrC}$ with the 20 % of $\sigma_{abs,365,extrapolate}$ and the $\sigma_{abs,470,BrC}^{cal}$ from homogeneous models. This method yields an underestimation of $AAE_{365/470,BrC}$; while its value can still reach 18, much higher than the upper limit of the AAE range reported in literature (Laskin et al., 2015), suggesting that the use of homogeneous models for RF10 may not be appropriate.

9. P18 L29. Please remove "in the meanwhile".

We thank the reviewer for the comment. We have removed "in the meanwhile" in P24 L10 in the marked-up version of the revised manuscript.

**References**

Ackerman, T. P. and Toon, O. B.: Absorption of visible radiation in atmosphere containing mixtures of absorbing and nonabsorbing particles, Appl. Opt., 20, 3661−3668, https://doi.org/10.1364/AO.20.003661, 1981.

Bahreini, R., Ervens, B., Middlebrook, A. M., Warneke, C., de Gouw, J. A., DeCarlo, P. F., Jimenez, J. L., Brock, C. A., Neuman, J. A., Ryerson, T. B., Stark, H., Atlas, E., Brioude, J., Fried, A., Holloway, J. S., Peischl, J., Richter, D., Walega, J., Weibring, P., Wollny, A. G., and Fehsenfeld, F. C.: Organic aerosol formation in urban and industrial plumes near Houston and Dallas, Texas, J. Geophys. Res. Atmospheres, 114, https://doi.org/10.1029/2008JD011493, 2009.

Chakrabarty, R. K., Moosmüller, H., Chen, L.-W. A., Lewis, K., Arnott, W. P., Mazzoleni, C., Dubey, M. K., Wold, C. E., Hao, W. M., and Kreidenweis, S. M.: Brown carbon in tar balls from smoldering biomass combustion, Atmospheric Chem. Phys., 10, 6363−6370, https://doi.org/10.5194/acp-10-6363-2010, 2010.

Dang, C., Segal-Rozenhaimer, M., Che, H., Zhang, L., Formenti, P., Taylor, J., Dobracki, A., Purdue, S., Wong, P.-S., Nenes, A., Sedlacek, A., Coe, H., Redemann, J., Zuidema, P., and Haywood, J.: Biomass burning and marine aerosol processing over the southeast Atlantic Ocean: A TEM single particle analysis, Atmospheric Chem. Phys. Discuss., 1−30, https://doi.org/10.5194/acp-2021-724, 2021.

Fierce, L., Riemer, N., and Bond, T. C.: Toward Reduced Representation of Mixing State for Simulating Aerosol Effects on Climate, Bull. Am. Meteorol. Soc., 98, 971−980, https://doi.org/10.1175/BAMS-D-16-0028.1, 2017.

Fischer, E. V., Jaffe, D. A., Marley, N. A., Gaffney, J. S., and Marchany-Rivera, A.: Optical properties of aged Asian aerosols observed over the U.S. Pacific Northwest, J. Geophys. Res. Atmospheres, 115, https://doi.org/10.1029/2010JD013943, 2010.

Formenti, P., Caquineau, S., Desboeufs, K., Klaver, A., Chevaillier, S., Journet, E., and Rajot, J. L.: Mapping the physico-chemical properties of mineral dust in western Africa: mineralogical composition, Atmospheric Chem. Phys., 14, 10663−10686, https://doi.org/10.5194/acp-14-10663-2014, 2014.

Hand, J. L., Malm, W. C., Laskin, A., Day, D., Lee, T., Wang, C., Carrico, C., Carrillo, J., Cowin, J. P., Collett Jr., J., and Iedema, M. J.: Optical, physical, and chemical properties of tar balls observed during the Yosemite Aerosol Characterization Study, J. Geophys. Res. Atmospheres, 110, https://doi.org/10.1029/2004JD005728, 2005.

Haywood, J. M., Abel, S. J., Barrett, P. A., Bellouin, N., Blyth, A., Bower, K. N., Brooks, M., Carslaw, K., Che, H., Coe, H., Cotterell, M. I., Crawford, I., Cui, Z., Davies, N., Dingley, B., Field, P., Formenti, P., Gordon, H., de Graaf, M., Herbert, R., Johnson, B., Jones, A. C., Langridge, J. M., Malavelle, F., Partridge, D. G., Peers, F., Redemann, J., Stier, P., Szpek, K., Taylor, J. W., Watson-Parris, D., Wood, R., Wu, H., and Zuidema, P.: Overview: The CLoud-Aerosol-Radiation Interaction and Forcing: Year-2017 (CLARIFY-2017) measurement

campaign, Atmospheric Chem. Phys. Discuss., 1–49, https://doi.org/10.5194/acp-2020-729, 2020.

He, C., Takano, Y., Liou, K.-N., Yang, P., Li, Q., and Mackowski, D. W.: Intercomparison of the GOS approach, superposition T-matrix method, and laboratory measurements for black carbon optical properties during aging, J. Quant. Spectrosc. Radiat. Transf., 184, 287–296, https://doi.org/10.1016/j.jqsrt.2016.08.004, 2016.

Hoffer, A., Gelencsér, A., Guyon, P., Kiss, G., Schmid, O., Frank, G. P., Artaxo, P., and Andreae, M. O.: Optical properties of humic-like substances (HULIS) in biomass-burning aerosols, Atmospheric Chem. Phys., 6, 3563–3570, https://doi.org/10.5194/acp-6-3563-2006, 2006.

Hoffer, A., Tóth, A., Nyirő-Kósa, I., Pósfai, M., and Gelencsér, A.: Light absorption properties of laboratory-generated tar ball particles, Atmospheric Chem. Phys., 16, 239–246, https://doi.org/10.5194/acp-16-239-2016, 2016.

Hu, D., Chen, J., Ye, X., Li, L., and Yang, X.: Hygroscopicity and evaporation of ammonium chloride and ammonium nitrate: Relative humidity and size effects on the growth factor, Atmos. Environ., 45, 2349–2355, https://doi.org/10.1016/j.atmosenv.2011.02.024, 2011.

Ito, A., Lin, G., and Penner, J. E.: Radiative forcing by light-absorbing aerosols of pyrogenetic iron oxides, Sci. Rep., 8, 7347, https://doi.org/10.1038/s41598-018-25756-3, 2018.

Kasthuriarachchi, N. Y., Rivellini, L.-H., Adam, M. G., and Lee, A. K. Y.: Light Absorbing Properties of Primary and Secondary Brown Carbon in a Tropical Urban Environment, Environ. Sci. Technol., 54, 10808–10819, https://doi.org/10.1021/acs.est.0c02414, 2020.

Kirchstetter, T. W., Novakov, T., and Hobbs, P. V.: Evidence that the spectral dependence of light absorption by aerosols is affected by organic carbon, J. Geophys. Res. Atmospheres, 109, https://doi.org/10.1029/2004JD004999, 2004.

Kurisu, M., Adachi, K., Sakata, K., and Takahashi, Y.: Stable Isotope Ratios of Combustion Iron Produced by Evaporation in a Steel Plant, ACS Earth Space Chem., 3, 588–598, https://doi.org/10.1021/acsearthspacechem.8b00171, 2019.

Laskin, A., Laskin, J., and Nizkorodov, S. A.: Chemistry of Atmospheric Brown Carbon, Chem. Rev., 115, 4335–4382, https://doi.org/10.1021/cr5006167, 2015.

Liati, A., Pandurangi, S. S., Boulouchos, K., Schreiber, D., and Arroyo Rojas Dasilva, Y.: Metal nanoparticles in diesel exhaust derived by in-cylinder melting of detached engine fragments, Atmos. Environ., 101, 34–40, https://doi.org/10.1016/j.atmosenv.2014.11.014, 2015a.

Liati, A., Pandurangi, S. S., Boulouchos, K., Schreiber, D., and Arroyo Rojas Dasilva, Y.: Metal nanoparticles in diesel exhaust derived by in-cylinder melting of detached engine fragments, Atmos. Environ., 101, 34–40, https://doi.org/10.1016/j.atmosenv.2014.11.014, 2015b.

Liu, C., Chung, C. E., Yin, Y., and Schnaiter, M.: The absorption Ångström exponent of black carbon: from numerical aspects, Atmospheric Chem. Phys., 18, 6259–6273, https://doi.org/10.5194/acp-18-6259-2018, 2018.

Liu, D., Taylor, J. W., Young, D. E., Flynn, M. J., Coe, H., and Allan, J. D.: The effect of complex black carbon microphysics on the determination of the optical properties of brown carbon, Geophys. Res. Lett., 42, 613–619, https://doi.org/10.1002/2014GL062443, 2015.

Liu, D., Whitehead, J., Alfarra, M. R., Reyes-Villegas, E., Spracklen, D. V., Reddington, C. L., Kong, S., Williams, P. I., Ting, Y.-C., Haslett, S., Taylor, J. W., Flynn, M. J., Morgan, W. T., McFiggans, G., Coe, H., and Allan, J. D.: Black-carbon absorption enhancement in the atmosphere determined by particle mixing state, Nat. Geosci., 10, 184–188, https://doi.org/10.1038/ngeo2901, 2017.

Liu, D., Li, S., Hu, D., Kong, S., Cheng, Y., Wu, Y., Ding, S., Hu, K., Zheng, S., Yan, Q., Zheng, H., Zhao, D., Tian, P., Ye, J., Huang, M., and Ding, D.: Evolution of Aerosol Optical Properties from Wood Smoke in Real Atmosphere Influenced by Burning Phase and Solar Radiation, Environ. Sci. Technol., 55, 5677–5688, https://doi.org/10.1021/acs.est.0c07569, 2021a.

Liu, Q., Liu, D., Wu, Y., Bi, K., Gao, W., Tian, P., Zhao, D., Li, S., Yu, C., Tang, G., Wu, Y., Hu, K., Ding, S., Gao, Q., Wang, F., Kong, S., He, H., Huang, M., and Ding, D.: Reduced volatility of aerosols from surface emissions to the top of the planetary boundary layer, Atmospheric Chem. Phys., 21, 14749–14760, https://doi.org/10.5194/acp-21-14749-2021, 2021b.

Machemer, S. D.: Characterization of Airborne and Bulk Particulate from Iron and Steel Manufacturing Facilities, Environ. Sci. Technol., 38, 381–389, https://doi.org/10.1021/es020897v, 2004.

Mackowski, D. W. and Mishchenko, M. I.: A multiple sphere T-matrix Fortran code for use on parallel computer clusters, J. Quant. Spectrosc. Radiat. Transf., 112, 2182–2192, https://doi.org/10.1016/j.jqsrt.2011.02.019, 2011.

Meter, S. L., Formenti, P., Piketh, S. J., Annegarn, H. J., and Kneen, M. A.: PIXE investigation of aerosol composition over the Zambian Copperbelt, Nucl. Instrum. Methods Phys. Res. Sect. B Beam Interact. Mater. At., 150, 433–438, https://doi.org/10.1016/S0168-583X(98)01020-9, 1999.

Moteki, N., Adachi, K., Ohata, S., Yoshida, A., Harigaya, T., Koike, M., and Kondo, Y.: Anthropogenic iron oxide aerosols enhance atmospheric heating, Nat. Commun., 8, 15329, https://doi.org/10.1038/ncomms15329, 2017.

Puffer, J. H., Russell, E. W. B., and Rampino, M. R.: Distribution and origin of magnetite spherules in air, waters, and sediments of the greater New York City area and the North Atlantic ocean, J. Sediment. Res., 50, 247–256, https://doi.org/10.1306/212F79BE-2B24-11D7-8648000102C1865D, 1980.

Saleh, R., Hennigan, C. J., McMeeking, G. R., Chuang, W. K., Robinson, E. S., Coe, H., Donahue, N. M., and Robinson, A. L.: Absorptivity of brown carbon in fresh and photo-chemically aged biomass-burning emissions, Atmospheric Chem. Phys., 13, 7683–7693, https://doi.org/10.5194/acp-13-7683-2013, 2013.

Saleh, R., Cheng, Z., and Atwi, K.: The Brown−Black Continuum of Light-Absorbing Combustion Aerosols, Environ. Sci. Technol. Lett., 5, 508–513, https://doi.org/10.1021/acs.estlett.8b00305, 2018.

Scarnato, B. V., Vahidinia, S., Richard, D. T., and Kirchstetter, T. W.: Effects of internal mixing and aggregate morphology on optical properties of black carbon using a discrete dipole approximation model, Atmospheric Chem. Phys., 13, 5089–5101, https://doi.org/10.5194/acp-13-5089-2013, 2013.

Sedlacek III, A. J., Buseck, P. R., Adachi, K., Onasch, T. B., Springston, S. R., and Kleinman, L.: Formation and evolution of tar balls from northwestern US wildfires, Atmospheric Chem. Phys., 18, 11289–11301, https://doi.org/10.5194/acp-18-11289-2018, 2018.

Shengo, M. L., Kime, M.-B., Mambwe, M. P., and Nyembo, T. K.: A review of the beneficiation of copper-cobalt-bearing minerals in the Democratic Republic of Congo, J. Sustain. Min., 18, 226–246, https://doi.org/10.1016/j.jsm.2019.08.001, 2019.

Sikamo, J., Mwanza, A., and Mweemba, C.: Copper mining in Zambia - history and future, J. South. Afr. Inst. Min. Metall., 116, 491–496, https://doi.org/10.17159/2411-9717/2016/v116n6a1, 2016.

Sinha, P., Hobbs, P. V., Yokelson, R. J., Bertschi, I. T., Blake, D. R., Simpson, I. J., Gao, S., Kirchstetter, T. W., and Novakov, T.: Emissions of trace gases and particles from savanna fires in southern Africa, J. Geophys. Res. Atmospheres, 108, https://doi.org/10.1029/2002JD002325, 2003.

Taylor, J. W., Wu, H., Szpek, K., Bower, K., Crawford, I., Flynn, M. J., Williams, P. I., Dorsey, J., Langridge, J. M., Cotterell, M. I., Fox, C., Davies, N. W., Haywood, J. M., and Coe, H.: Absorption closure in highly aged biomass burning smoke, Atmospheric Chem. Phys., 20, 11201–11221, https://doi.org/10.5194/acp-20-11201-2020, 2020.

Till, J. L., Guyodo, Y., Lagroix, F., Morin, G., and Ona-Nguema, G.: Goethite as a potential source of magnetic nanoparticles in sediments, Geology, 43, 75–78, https://doi.org/10.1130/G36186.1, 2015.

Vítková, M., Ettler, V., Johan, Z., Kříbek, B., Šebek, O., and Mihaljevič, M.: Primary and secondary phases in copper-cobalt smelting slags from the Copperbelt Province, Zambia, Mineral. Mag., 74, 581–600, https://doi.org/10.1180/minmag.2010.074.4.581, 2010.

Zhang, H., Zhou, C., Wang, Z., Zhao, S., and Li, J.: The influence of different black carbon and sulfate mixing methods on their optical and radiative properties, J. Quant. Spectrosc. Radiat. Transf., 161, 105–116, https://doi.org/10.1016/j.jqsrt.2015.04.002, 2015a.

Zhang, X. L., Wu, G. J., Zhang, C. L., Xu, T. L., and Zhou, Q. Q.: What is the real role of iron oxides in the optical properties of dust aerosols?, Atmospheric Chem. Phys., 15, 12159–12177, https://doi.org/10.5194/acp-15-12159-2015, 2015b.